# Time-resolved proteomic analyses of senescence highlight metabolic rewiring of mitochondria

Jun Yong Kim[1] , Ilian Atanassov[1] , Frederik Dethloff[1], Lara Kroczek[1] , Thomas Langer[1,2]

**Mitochondrial dysfunction and cellular senescence are hallmarks of aging. However, the relationship between these two phenomena remains incompletely understood. In this study, we investigated the rewiring of mitochondria upon development of the senescent state in human IMR90 fibroblasts. Determining the bioenergetic activities and abundance of mitochondria, we demonstrate that senescent cells accumulate mitochondria with reduced OXPHOS activity, resulting in an overall increase of mitochondrial activities in senescent cells. Time-resolved proteomic analyses revealed extensive reprogramming of the mitochondrial proteome upon senescence development and allowed the identification of metabolic pathways that are rewired with different kinetics upon establishment of the senescent state. Among the early responding pathways, the degradation of branched-chain amino acid was increased, whereas the one carbon folate metabolism was decreased. Late-responding pathways include lipid metabolism and mitochondrial translation. These signatures were confirmed by metabolic flux analyses, highlighting metabolic rewiring as a central feature of mitochondria in cellular senescence. Together, our data provide a comprehensive view on the changes in mitochondrial proteome in senescent cells and reveal how the mitochondrial metabolism is rewired in senescent cells.**

## Introduction

Cellular senescence (CS) is known to contribute to a wide array of age-related diseases such as cancer, cardiovascular diseases, and osteoarthritis (1). Diverse stressors including genotoxic, epigenotoxic, oxidative, and oncogenic insults induce the senescent state of cells, which is characterized by the secretion of a plethora of bioactive molecules, termed senescence-associated secretory phenotype (SASP) (2). The SASP mainly comprises pro-inflammatory cytokines, growth factors, and extracellular matrix modifiers, which remodel the tissue environment of senescent cells. It largely depends on the type of stress, the type of the recipient cell, and the duration of being senescent. Thus, both the composition and the temporal dynamics of the SASP determine how senescent cells affect their environment. For example, acute SASP is necessary for tissue development and wound healing (3, 4, 5, 6, 7), whereas chronic SASP is detrimental and disrupts tissue homeostasis, driving age-related dysfunctions and diseases (8). Accordingly, there has been great interest either in eliminating senescent cells or modulating the chronic SASP to tackle age-related diseases, called senotherapy (9).

Mitochondria have been shown to play regulatory roles in CS and modulate the SASP. Increased mitochondrial biogenesis and decreased turnover of mitochondria by mitophagy result in the accumulation of mitochondria in senescent cells (10). Correlating with the abundance of mitochondria, increased mitochondria-derived reactive oxygen species (mtROS) potentiate the DNA damage response in senescent cells (11) and enhance the SASP by promoting the formation of cytoplasmic chromatin fragments, which activate innate immune signaling along the cGAS–STING pathway (12). Moreover, oxidative phosphorylation (OXPHOS) regulates CS and modulates the SASP. Senescent cells are characterized by a higher mitochondrial fatty acid oxidation (FAO) and the inhibition of FAO led to an impaired SASP expression (13). Increased activity of the pyruvate dehydrogenase complex, which converts pyruvate to acetyl-CoA in mitochondria, enhances OXPHOS activity and is a rate-limiting factor to drive oncogene-induced senescence (14). Similarly, increased OXPHOS activity in senescent cells governs the strength of the SASP by promoting $NAD^+$ regeneration, preventing the activation of AMPK-p53 signaling, which is known to suppress the SASP (15). On the other hand, a decreased cellular $NAD^+/NADH$ ratio upon OXPHOS dysfunction is sufficient to drive cells into senescence but results in a distinct SASP profile lacking pro-inflammatory IL1 cytokines (16). Together, these studies establish a central role of mitochondria in CS and the SASP and posit mitochondria as an attractive target for senotherapy (17).

Although the importance of mitochondria for CS and the SASP has been established, the functional state of mitochondria in senescent cells remained unclear. Several studies reported an OXPHOS dysfunction and lower mitochondrial membrane potential (MMP) in senescent cells (18, 19, 20, 21, 22, 23), whereas the MMP was

[1]Max Planck Institute for Biology of Ageing, Cologne, Germany    [2]Cologne Excellence Cluster on Cellular Stress Responses in Aging-Associated Diseases (CECAD), University of Cologne, Cologne, Germany

Correspondence: tlanger@age.mpg.de
Jun Yong Kim's present address is Department of Cell Biology, Blavatnik Institute, Harvard Medical School, Boston, MA, USA

found to increase with mitochondrial abundance in these cells (24). Moreover, the increased catabolism of central carbons such as pyruvate, fatty acids, and glutamine in senescent cells is difficult to reconcile with dysfunctional mitochondria (13, 14, 15, 25, 26 *Preprint*, 27).

In this study, we performed an in-depth, time-resolved analysis of the mitochondrial proteome upon the establishment of CS. Our findings discover the metabolic rewiring as a central feature of mitochondria in senescent cells and define the functional state of mitochondria in senescent cells, which provides a possible explanation for apparent discrepancies in the literature.

# Result

## Accumulation of mitochondria with reduced bioenergetic activity in senescent fibroblasts

We treated IMR90 human lung fibroblasts with two chemotherapeutic agents, decitabine and doxorubicin to establish CS. Decitabine is a deoxycytidine analog harboring nitrogen instead of a carbon atom at the 5′ position of the pyrimidine ring (Fig S1A). Upon incorporation into the replicating genomic DNA, it impairs DNA methylation and causes epigenetic stress. Doxorubicin, on the other hand, blocks topoisomerase II and causes DNA damage. Treatment of IMR90 fibroblasts with decitabine or doxorubicin for 7 d increased mRNA levels of CDKN1A, decreased mRNA levels of LMNB1, and induced the common SASP genes IL1A and IL6, indicating the senescent state (Fig S1B and C). To unambiguously demonstrate that the decitabine-induced senescent features are irreversible as per the definition of CS, we removed decitabine from senescent cells on day 7 and monitored cell-cycle activity, senescence-associated beta-galactosidase (SA-$\beta$-Gal) activity, and senescence-associated protein markers after cultivating cells for another week. Cells rapidly decreased proliferation and maintained the proliferation-deficient state after decitabine washout (Fig S1D). SA-$\beta$-Gal positivity was maintained after 7 d of decitabine removal (Fig S1E). Similarly, loss of phosphorylated Rb at serine 807/811 (p-Rb$^{S807/811}$) and cyclin A2 (CCNA2), which are essential for G1-S transition, indicated an irreversible cell cycle arrest (Fig S1E). DNA damage was acutely induced as evidenced by the accumulation of phosphorylated H2A.X at serine 139 (p-H2A.X$^{S139}$), whereas the level of lamin B1 (LMNB1), a hallmark of senescence (28), were reduced (Fig S1F). We conclude from these experiments that IMR90 fibroblasts attained a senescent state upon treatment with the chemotherapeutic agents.

To examine mitochondrial functions, we first determined the mitochondrial abundance in these cells, examining the volume of mitochondria, rather than relying on a two-dimensional analysis of the mitochondrial network. We used an immunocytochemistry-based quantification method, which, in contrast to other probes such as MitoTracker and nonyl acridine orange, allowed the determination of the mitochondrial volume largely independent of mitochondrial activities (29). Confocal images of the mitochondria from a single cell were stacked and rendered into a three-dimensional image using MitoGraph 3.0 (Fig 1A) (30). It showed that

the mitochondrial length was increased on average around eightfold in a senescent fibroblast, whereas the average width remained unaltered (Fig 1B). As a result, the volume of the mitochondria was increased around eightfold in a senescent fibroblast when compared with a proliferating fibroblast. This corresponds to the eightfold increase in the volume of a senescent fibroblast (31), indicating that the mitochondrial volume increases proportionally with the cellular volume in senescent cells.

We next determined mitochondrial DNA (mtDNA) levels in senescent IMR90 fibroblasts. Although the relative amount of mtDNA was higher in these cells, normalization to the mitochondrial volume revealed a proportional increase of mtDNA with the mitochondrial volume (Fig 2A). Senescent fibroblasts showed an overall higher MMP, which however is decreased when normalized to the mitochondrial volume (Fig 2B and C). We observed a similar pattern for the amount of polarized mitochondria and mitochondrial superoxide levels in the senescent fibroblasts (Fig 2D–F). In agreement with the alterations in the MMP, senescent cells showed an increased oxygen consumption rate (OCR) on a cellular basis, which however corresponds to a decreased OCR per mitochondrial volume (Fig 2G and H).

We therefore conclude that the bioenergetic activity of mitochondria is decreased in senescent fibroblasts. However, the accumulation of mitochondria causes an overall enhancement of mitochondrial bioenergetic parameters in these cells. These findings highlight the importance to consider mitochondrial abundance when assessing the functional status of mitochondria and possibly resolve conflicting reports on mitochondrial fitness in senescent cells.

## Reshaping of the mitochondrial proteome during CS development

To gain further insights into the reprogramming of mitochondria in senescence, we analyzed the mitochondrial proteome upon transition to the senescent state by tandem mass tag labeling mass spectrometry on days 1, 3, 5, and 7 after the treatment with DMSO or decitabine (Fig 3A). 6,482 proteins were quantified in all samples, corresponding to more than 80% of the total identified proteins, and used for further analyses (Fig S2A). Principal component analysis revealed progressive changes in the cellular proteome during the development of the CS, whereas that of proliferating control cells with DMSO remained largely unaltered as expected (Fig S2B). Changes in the protein levels of key senescence markers confirmed the establishment of the CS by decitabine (Fig S2C). Our dataset covered around 60% of the proteomes of major cellular organelles based on the reference proteome of each organelle, including the nucleus (32), cytosol (33), ER membrane (34), and mitochondria (35) (Fig S2D). Please see the Supplemental Data 1 as a source data for the proteomic analyses.

To define mitochondrial proteomic changes, we first examined whether the increased mitochondrial abundance in senescent cells introduces bias in our proteomic analysis. We calculated the proportion of mitochondrial proteins in the cellular proteome at each time point but did not observe alterations in the fraction of mitochondrial proteins during the CS development, unlike that of the ER membrane or nuclear proteins (Figs 3B and S2F). We also compared mitochondrial proteomic changes by two different

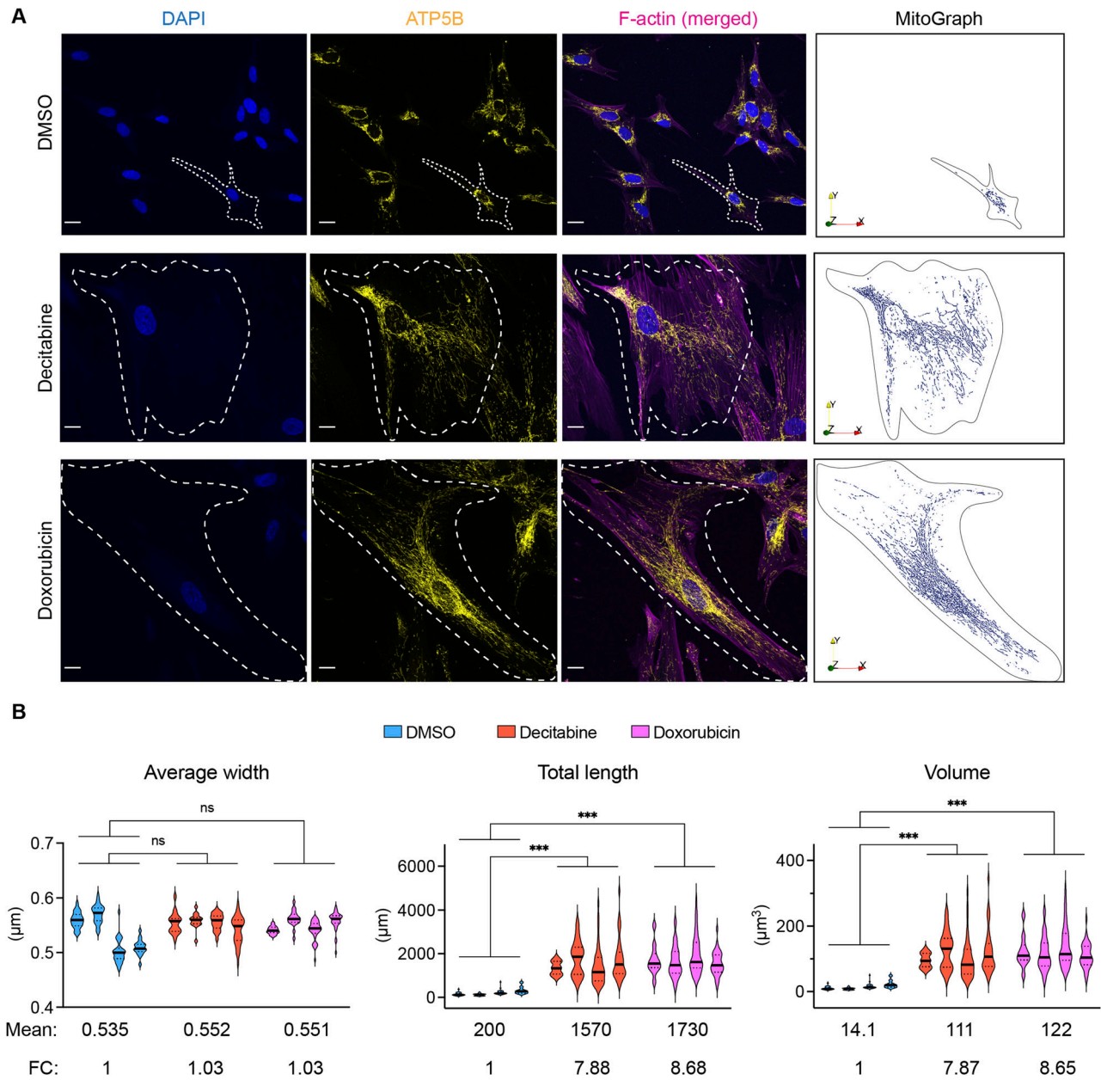

**Figure 1. Determination of the mitochondrial amount in senescent fibroblasts.**
**(A)** Representative images of the mitochondria in IMR90 fibroblasts on day 7 after the treatment with DMSO, decitabine or doxorubicin. (Left) The maximal projection of confocal images from different z-levels is shown. (Right) Each z-stack image was combined and rendered into a three-dimensional image using MitoGraph 3.0 (30). ATP5B was used as a mitochondrial marker. F-actin was stained to define a cellular boundary. **(B)** Quantification of mitochondrial length, width, and volume using MitoGraph 3.0. Data are presented as violin plots. Median: bold line, quartiles: dotted lines. Between 11 and 44 cells were analyzed per replicate and condition. The average mean and fold changes are shown in the graph. FC: fold change. Nested one-way ANOVA, Dunnett correction. n = 4.

normalization units: total peptide counts and mitochondrial-specific peptide counts. The comparison yielded extremely high correlations between fold changes calculated by the two normalization units at all time points (Fig S2E). These results indicate that the mitochondrial proteome increased proportionately to the cellular proteome throughout the development of the CS and the proteomic changes can be faithfully analyzed from the data normalized by the total peptide counts.

The mitochondrial proteome was significantly altered during the CS development, yielding 279 differentially expressed genes (DEGs) on day 7, corresponding to nearly 40% of the total mitochondrial proteins quantified (Fig 3C and D). We observed similar changes for nuclear, cytosolic, and ER membrane proteins, which indicate a lack of strong bias in organellar proteomic changes during the CS development (Fig 3D). Based on the affected mitochondrial pathways (MitoPathways, curated in MitoCarta 3.0), the 279 mitochondrial

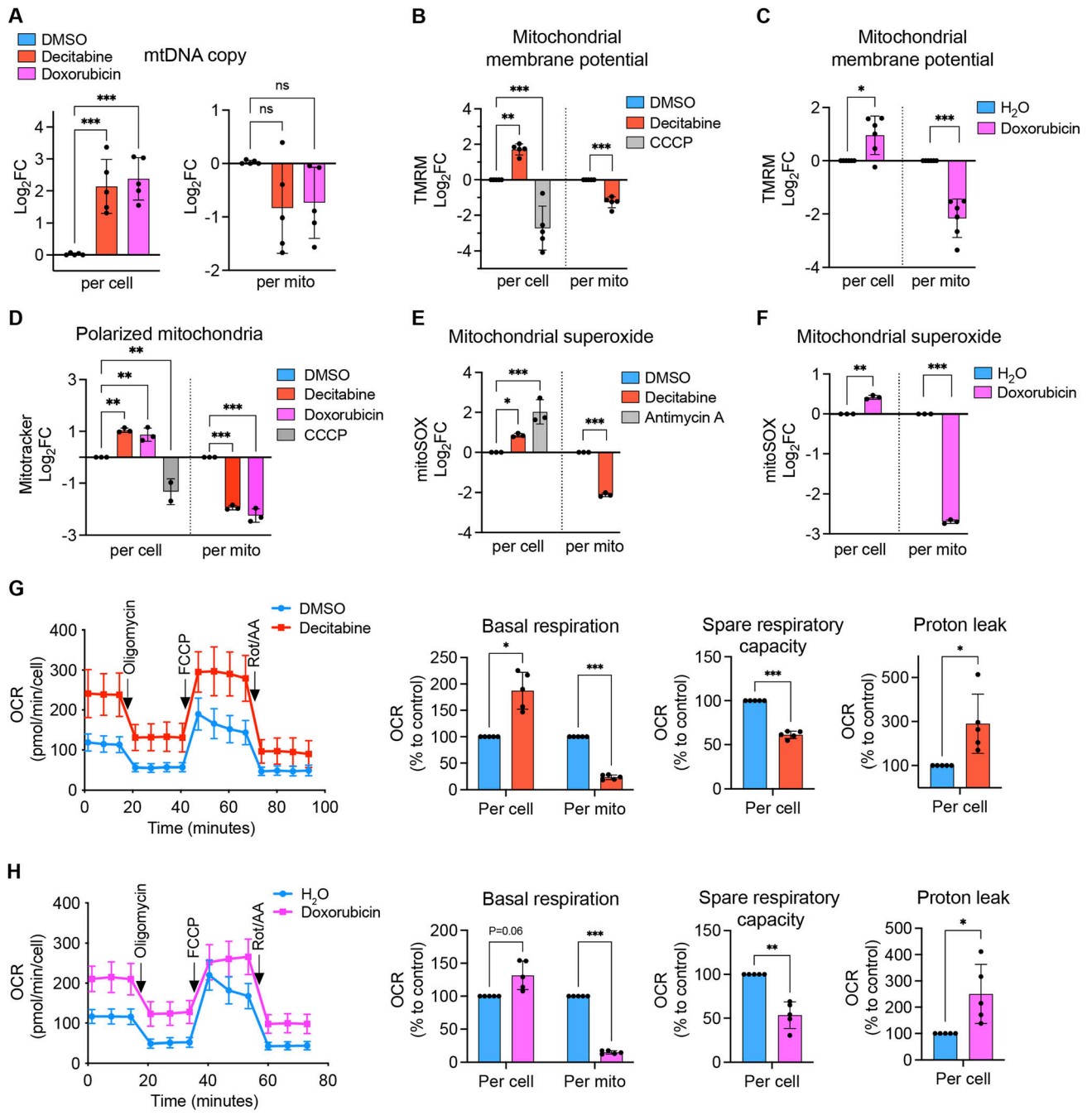

**Figure 2. Accumulation of mitochondria with reduced bioenergetic activity in senescent fibroblasts.**
The values are first measured per cellular basis on day 7 after the treatment with DMSO, decitabine or doxorubicin to IMR90 fibroblasts and followed by scaling with the relative mitochondrial volume per cell. **(A)** Determination of mtDNA levels in cellular DNA extracts by qRT–PCR using MT-ND1 and ACTB as probes for mtDNA and nuclear DNA, respectively. Mean ± SD. One-way ANOVA, Dunnett correction. n = 5 from independent cultures. **(B, C, D, E, F)** Measurement of mitochondrial membrane potential, amount of polarized mitochondria, and mitochondrial superoxide levels. IMR90 fibroblasts were stained with TMRM (B, C), Mitotracker Deep Red FM (D), or mitoSOX (E, F) and analyzed by flow cytometry. Antimycin A (10 $\mu$M) and CCCP (50 $\mu$M) were used as positive controls. Mean ± SD. **(B, D, E)**: one-way ANOVA, Dunnett correction. **(C, F)**: Welch $t$ test, Bonferroni–Dunn correction. (B): n = 5, (C): n = 6, (D, E, F): n = 3 from independent cultures. **(G, H)** Measurement of oxygen consumption rate. Left: real-time oxygen consumption rate before and after the sequential addition of oligomycin (1 $\mu$M), FCCP (0.5 $\mu$M), and rotenone/antimycin A (Rot/AA, 0.5 $\mu$M each). Right: fold changes calculated from the left graphs. The % values were transformed to Log$_2$ values and subjected to statistical analysis. Mean ± SD. Welch $t$ test, Bonferroni–Dunn correction. n = 5.

DEGs on day 7 were categorized into 6 major groups and the percentage of DEGs within each group was calculated. This analysis showed a general up-regulation of genes related to metabolism, signaling, dynamics/surveillance, and down-regulation of mtDNA-related genes, whereas we observed mixed alterations in genes related to OXPHOS and mitochondrial proteostasis (Fig 3E).

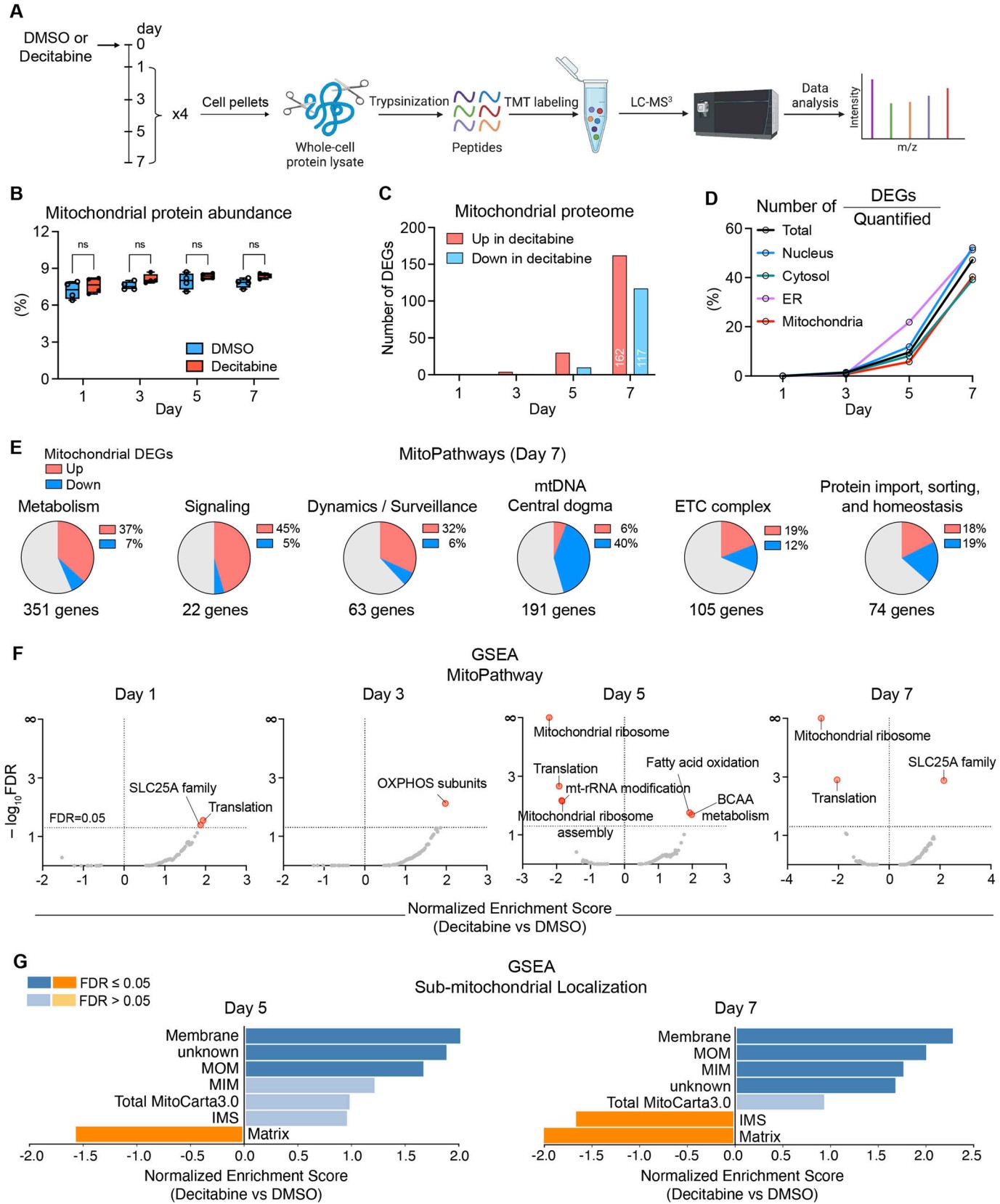

**Figure 3. Reprogramming of the mitochondrial proteome upon the development of CS.**
**(A)** Workflow for the time-resolved analysis of the mitochondrial proteome upon CS induction by decitabine. Cellular proteome was measured by tandem mass tag labeling mass spectrometry on days 1, 3, 5, and 7 after DMSO or decitabine treatment in IMR90 fibroblasts. All samples were measured in biological quadruplicates. **(B)** The

Changes in the mitochondrial proteome were also subjected to a gene set enrichment analysis (GSEA), which highlighted alterations in metabolic pathways (e.g., branched-chain amino acid metabolism, FAO, SLC25A family) and in the translation of mtDNA-encoded genes (Fig 3F), corroborating the previous analysis (Fig 3E). Moreover, the GSEA of submitochondrial localization revealed a general increase in inner and outer membrane proteins, whereas matrix proteins were decreased on days 5 and 7 (Fig 3G). These alterations mainly result from a general increase in SLC25A family proteins which are integral membrane proteins and an overall decrease in the translational apparatus in the matrix space. The increase in membrane proteins is unlikely because of the enhanced protein import because the small TIM proteins (TIMM8B, TIMM9, TIMM10, TIMM13) which are responsible for the chaperone-mediated import of many hydrophobic membrane proteins were reduced altogether on day 7 (Fig S3; protein import & sorting). Considering that the mitochondrial protein abundance remained constant throughout the CS development, these analyses suggest remodeling of the mitochondrial proteome with an altered ratio between the membrane and matrix proteins.

### Changes in the mitochondrial proteome reveal broad metabolic rewiring of senescent fibroblasts

We further examined how individual pathways were affected throughout the CS establishment. We identified four groups of DEGs differing in their temporal dynamic patterns (Figs 4A and S4 for detailed lists of genes). Groups 1 and 2 include mitochondrial proteins that were altered rather early (day 3 or 5) and whose abundance either kept increasing (group 1) or decreasing (group 2) upon the decitabine treatment (Fig 4A). On the contrary, groups 3 and 4 contain late-responding proteins, whose abundance changed only on day 7. Notably, only 1.5% of the analyzed DEGs fluctuated over the CS development and did not fall into any of the four groups (Fig 4C).

Each group of proteins was then subjected to an overrepresentation analysis with KEGG and GO databases (Fig 4B). The analysis revealed an enrichment of branched-chain amino acid (BCAA) catabolism in group 1 and the mitochondrial arm of the one-carbon (1C)-folate metabolism in group 2 (Fig 4B). Proteins associated with fatty acid metabolism and calcium import into the mitochondria were overrepresented in the late-responding group 3, indicating up-regulation of these pathways upon senescence induction (Fig 4A and B), in agreement with previous findings (13, 36). On the contrary, we observed a strong enrichment of subunits of mitochondrial ribosomes and components of the mitochondrial gene expression apparatus in group 4, suggesting a reduction of

mitochondrial translation in the established senescent state (Figs 4A and B). To validate these findings and to exclude any bias because of mitochondrial abundance, we synthesized mtDNA-encoded proteins in isolated mitochondria in the presence of $^{35}$S-methionine. In agreement with our proteomic analysis, mitochondrial translation was reduced in the senescent fibroblasts (Fig 4D).

Together, we conclude that mitochondria are broadly rewired upon CS induction, pointing to metabolic adaptations, favoring the degradation of BCAA, and down-regulating the 1C-folate metabolism. This is accompanied by a decrease in mitochondrial translation, consistent with the observed decreased respiratory activity of mitochondria in senescent cells.

### Enhanced catabolism of BCAA in senescent fibroblasts

In further experiments, we used metabolic tracing experiments to validate early metabolic adaptations indicated by our proteomic analysis. We observed the accumulation of enzymes of the BCAA metabolism upon CS induction, suggesting an enhanced degradation of BCAAs in the senescent cells (Fig 5A). The nitrogen in BCAAs accumulates in glutamate, which is used to synthesize several nonessential amino acids (NEAAs). On the other hand, carbon atoms of BCAA are found in acyl-CoAs, used for the synthesis of fatty acids or cholesterol, or fed into the TCA cycle (Fig 5B). To monitor the catabolism of BCAA in senescent cells, we performed metabolic tracing experiments with BCAAs that are labeled with stable isotopes of either nitrogen or carbons. These experiments revealed an increased flux of both carbons and nitrogen to the downstream metabolites in the senescent fibroblasts (Fig 5C and D). We also observed an accumulation of BCAAs-derived short-chain acylcarnitines such as acetyl-carnitine, propionyl-carnitine, and isobutyryl-carnitine (Fig S5C), which points to enhanced BCAA degradation (37). These data demonstrate the validity of the proteomics signature of increased BCAA degradation in senescent cells.

To address the functional impact of the mitochondrial reprogramming of BCAA metabolism in senescent cells, we blocked BCAA catabolism by knocking down the rate-limiting enzyme BCKDHA, followed by a treatment of decitabine or doxorubicin to induce senescence (Fig S5D). The knockdown of BCKDHA caused significant cell deaths largely regardless of the senescence-inducing DNA damage. Therefore, BCAA catabolism through BCKDHA is essential in our cell culture conditions, hampering the assessment of the role of BCAA catabolism for the function of senescent cells.

---

percentage of mitochondrial proteins within the cellular proteome during CS development. Mitochondrial protein abundance was calculated as $2^{\Sigma(\text{mitochondrial peptides reporter intensities})}/2^{\Sigma(\text{total reporter intensities})}$. Whisker: mean, box ends: 25% and 75% quantiles. Welch $t$ test at each time point, Bonferroni–Dunn correction. n = 4. **(C)** The number of differentially expressed genes (DEGs) encoding mitochondrial proteins at each time point of CS development. **(D)** The percentage of DEGs encoding different organellar proteins among all quantified proteins at different time points of CS development. **(E)** Representation of mitochondrial pathways within DEGs on day 7. Six major categories can be distinguished using MitoPathways enlisted in the human MitoCarta 3.0. The number of genes beneath the circles indicates the number of quantified proteins in each category. **(F)** Gene set enrichment analysis of the proteomics data according to MitoPathways. FDR = 0.05 was used as the cutoff. **(G)** Gene set enrichment analysis of submitochondrial localization of the proteomics data on days 5 and 7 according to the human MitoCarta 3.0. MOM, mitochondrial outer membrane; MIM, mitochondrial inner membrane; IMS, intermembrane space.

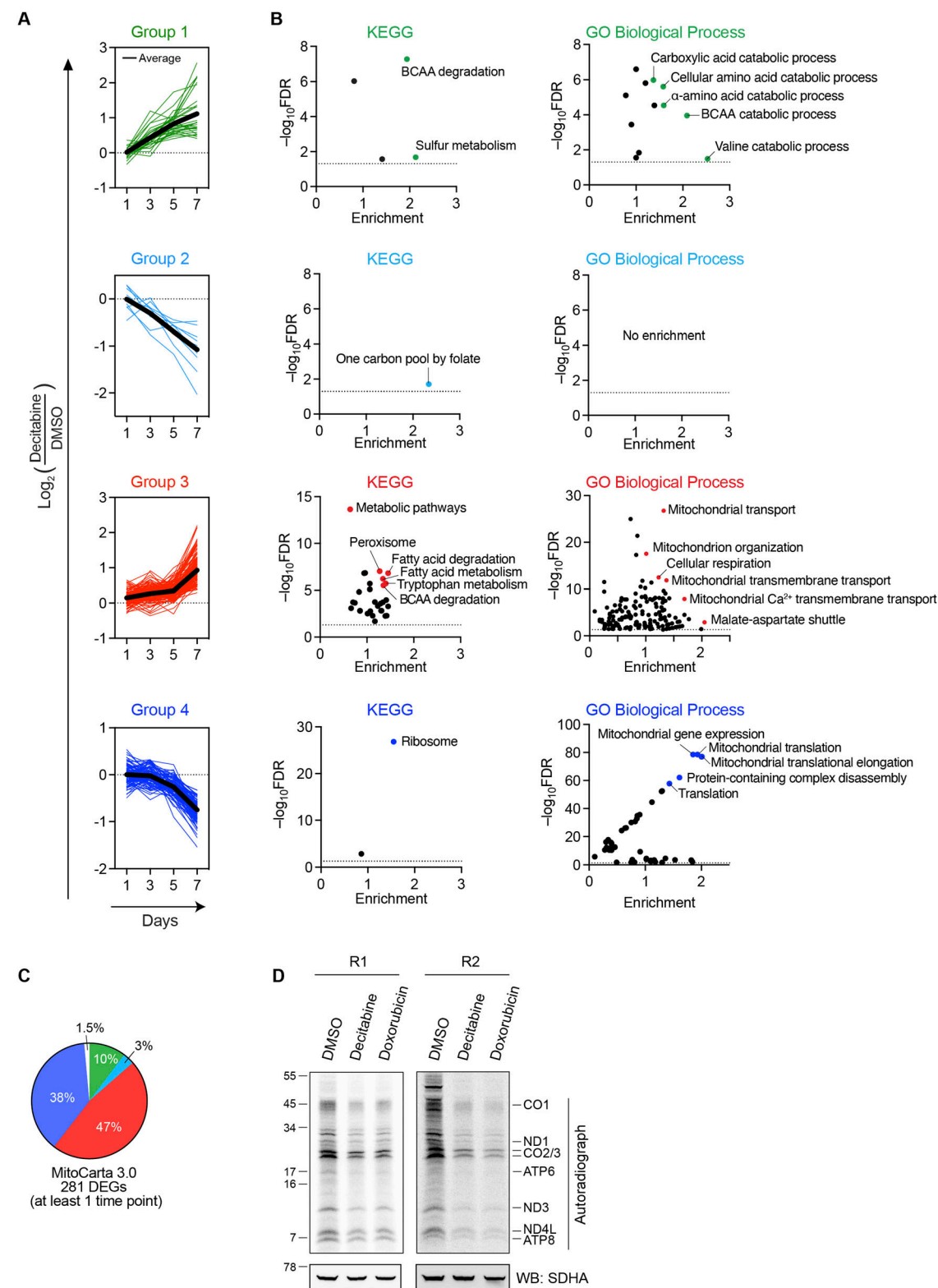

**Figure 4. Classification of mitochondrial proteins according to their time-dependent changes during the CS development.**
**(A)** Classification of mitochondrial DEGs into four groups according to the temporal dynamics of Log$_2$ fold changes during the development of CS. The bold line in each group indicates the average of Log$_2$ values. For detailed lists of genes in each category, see Fig S4. **(B)** Overrepresentation analysis of mitochondrial DEGs in each group from (A) based on two different databases. KEGG, Kyoto Encyclopedia of Genes and Genomes; GO, Gene Ontology. Highlighted are terms with the highest enrichments and/or significance. The color corresponds to the groups in (A). **(C)** Percentage of each group within total mitochondrial DEGs from (A). The color corresponds to the groups in (A). **(D)** In organello assay of mitochondrial translation. Mitochondria were isolated from IMR90 fibroblasts on day 7 after the treatment with indicated compounds and

### Early reduction of 1C-folate metabolism in senescent fibroblasts

Our proteomic analysis also suggested that the 1C-folate cycle is an early responding pathway that is rapidly reduced upon the decitabine treatment (Fig 4B; group 2, Fig S4A; group 2). Notably, although the overrepresentation analysis was restricted to mitochondrial proteins, enzymes involved in the cytosolic arm of 1C-folate metabolism were also acutely decreased upon induction of the senescent state (Fig 6A). To validate the proteomic footprints in 1C-folate metabolism, we performed targeted metabolomics, focusing on polar metabolites including nucleotides and amino acids. Principal component analysis showed that the metabolome of the senescent fibroblasts is distinct from that of proliferating cells (Fig S5A). The metabolomics revealed a significant reduction of purines (AMP, GMP) and deoxy-thymidylates (dTTP, note that dTMP was under the detection threshold exclusively in the senescent cells), which is consistent with the reduced 1C-folate metabolism (Figs 6B and S5B and C). Another indicator of the activity of the pathway is the serine catabolism by cytosolic SHMT1 and mitochondrial SHMT2. We found decreased glycine levels and an increased serine-to-glycine ratio in the senescent cells, consistent with the decreased SHMT2 level in these cells (Fig 6C and D). These observations were further substantiated by tracing carbons of glucose, which demonstrated that the formation of serine from glucose and glycine from serine was significantly reduced (Fig 6E). To distinguish effects on the cytosolic and mitochondrial arms of the 1C-folate metabolism, we performed tracing experiments using a serine stable isotope with deuterium (Fig 6F). Monitoring the accumulation of dTTP isotopologues allowed us to determine the directionality of the pathway (38). M+1 dTTP was exclusively detected but not M+2 dTTP (Fig 6G), indicating that serine was catabolized exclusively in the mitochondria in proliferating IMR90 fibroblasts. Accordingly, inhibition of the 1C-folate/serine catabolism in senescent cells results in the depletion of deoxy-thymidylates in the senescent cells (Fig 6B), without any detectable increase in the cytosolic catalysis of serine (Fig 6G). Thus, the 1C-folate metabolism is down-regulated in the senescent cells in accordance with the proteomic analysis.

## Discussion

We have performed a time-resolved proteomic analysis to define mitochondrial adaptations in senescent cells. Using anticancer drugs in human fibroblasts as a CS model (39), we show broad reshaping of the mitochondrial proteome and metabolic rewiring of mitochondria upon establishment of the senescent state. About 40% of the mitochondrial proteins were significantly changed in senescent cells, with membrane proteins being generally enriched over soluble matrix proteins. Our time-resolved proteomic analysis did not only establish a broad rewiring of the mitochondrial proteome but also allowed us to distinguish early and late proteomic adaptations. Our enrichment analysis yielded primarily metabolism-related signatures among the early responding pathways, highlighting the importance of metabolic rewiring of mitochondria in senescence.

We identified enhanced catabolism of BCAAs as an early responding metabolic pathway in senescent cells. We observed an increased flux of both nitrogen and carbons of BCAAs to their downstream metabolites, such as some NEAAs and acyl-CoAs. On one hand, transamination of BCAAs to the NEAAs supports the maintenance of the levels of alanine, glutamate, proline, and serine, which we found to be preserved in senescent cells. Acyl-CoAs, on the other hand, are used for lipid synthesis. Of note, despite the increase in BCAA catabolism, there was no significant incorporation of the BCAA carbons into TCA cycle (malate). Rather, we found three to five times more flux into the lipogenic TCA cycle intermediates such as acetyl-CoA and citrate. Moreover, in agreement with previous findings showing that senescent cells enhance both synthesis and oxidation of fatty acids (13, 40, 41, 42, 43), we observed a late up-regulation of lipid metabolizing proteins in our proteomic analysis. It is therefore conceivable that BCAAs serve as a source for lipid synthesis in senescent cells rather than being fully oxidized to support mitochondrial respiration, as has been described for adipose tissue (44, 45).

In contrast to BCAA degradation, the 1C-folate metabolism was down-regulated early in CS development. The 1C-folate cycle provides intermediates for the synthesis of purines and dTMP in the cytosol which are required for the replication of the genome. Low demand for nucleotide synthesis with the reduced 1C-folate metabolism is therefore consistent with the stable cell cycle arrest of senescent cells. Indeed, inhibition of deoxynucleotide metabolism was found to be both necessary and sufficient for oncogene-induced senescence (46). In addition to its role in nucleotide synthesis, the 1C-folate cycle supports mitochondrial translation by supplying formyl-methionine for translation initiation in mitochondria (47). Our metabolic tracing experiments revealed that serine is catabolized exclusively via the mitochondrial arm of the 1C-folate cycle, which is down-regulated in senescent cells without compensation from the cytosol. Accordingly, we observed reduced synthesis of mitochondrially encoded OXPHOS subunits in senescent cells, consistent with the decreased bioenergetic activity of the mitochondria in these cells. Because a decrease in mitochondrial translation and OXPHOS activity makes cells vulnerable to inhibition of glycolysis (47), our data would explain the susceptibility of senescent cells to glucose restriction (27). It should be noted that mitochondrial translation is a late-responding pathway upon induction of the senescent state. This likely explains why we did not observe a general decrease of OXPHOS subunits in our proteomic analysis, despite attenuated mitochondrial translation.

The mitochondrial volume was increased around eightfold on average in senescent IMR90 fibroblasts. The increased mitochondrial volume in senescent cells can be attributed to both enhanced

incubated in a translation buffer in the presence of $^{35}$S-methionine as described in the Materials and Methods section. Each protein was annotated based on direct verification by immunoblot and size information. SDHA blot was used as a reference for equal loading. Two representative autoradiographs are shown.

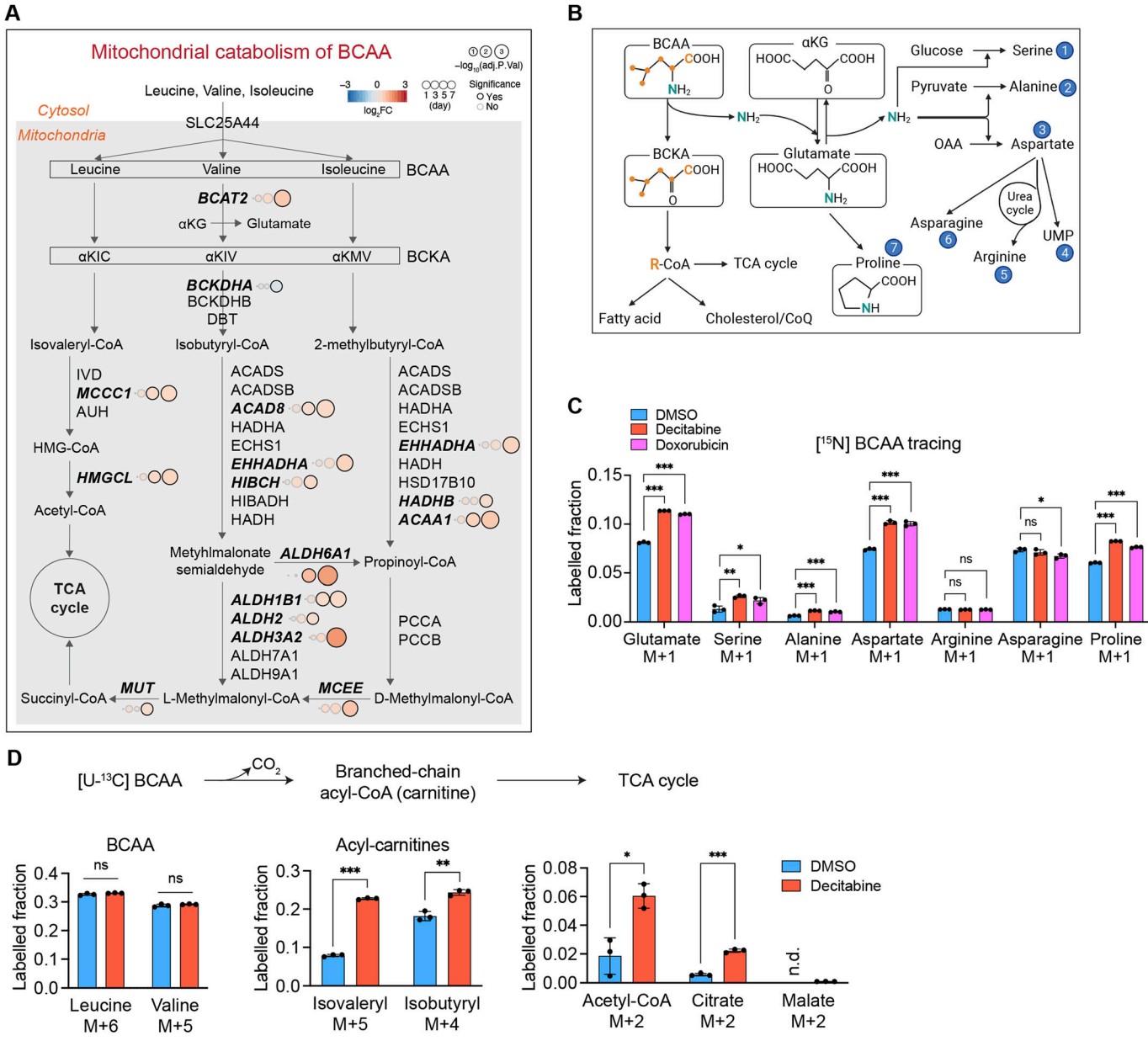

**Figure 5. Enhanced mitochondrial BCAA degradation in senescent fibroblasts.**
**(A)** The BCAA catabolism in mitochondria. Enzyme levels are derived from the proteomics data. The DEGs are shown as closed circles and in a bold italicized font. The color code indicates Log₂ fold changes. BCKA, branched-chain α-keto acid; αKIC, α-keto-isocaproate; αKIV, α-keto-isovalerate; αKMV, α-keto-beta-methylvalerate. **(B)** Carbon and nitrogen flux throughout BCAA catabolism. Metabolites incorporating BCAA-derived nitrogen are marked with circled numbers and highlighted in blue. αKG, α-ketoglutarate; CoQ, coenzyme Q; OAA, oxaloacetate. **(C)** Metabolic tracing of the BCAA metabolism using ¹⁵N-L-leucine and ¹⁵N-L-valine at equimolar concentrations (100 μM each) for 24 h in IMR90 fibroblasts on day 7 after the treatment of indicated compounds. Mean ± SD. One-way ANOVA for each metabolite, Dunnett correction. n = 3 from independent cultures. **(D)** Metabolic tracing of the BCAA metabolism using ¹³C₆-L-leucine and ¹³C₅-L-valine at equimolar concentrations (100 μM each) for 2.5 h in IMR90 fibroblasts on day 7 after the treatment of the indicated compounds. n.d.: not detected. Mean ± SD. Welch t test, Bonferroni–Dunn correction. n = 3 from independent cultures.

mitochondrial biogenesis and impaired mitophagy (10). For example, the mitochondrial biogenesis factor PGC-1β mediates the increase of mitochondrial abundance in senescent cells (11). On the other hand, mitophagy was shown to be impaired in these cells (48), which also exhibit lysosomal dysfunctions (49). However, it is important to note that despite the larger mitochondrial volume, the mitochondrial proteome scaled with the cellular proteome in senescent cells which are about 2.5-fold more in protein mass (Fig S1H) and eightfold larger in volume compared with proliferating cells (31). This contrasts with the nuclear proteome whose fraction on the cellular proteome decreased, and the ER proteome, whose fraction increased in senescent cells, in agreement with a previous finding (50). The changes in mitochondrial volume upon establishment of the senescent state must be taken into account

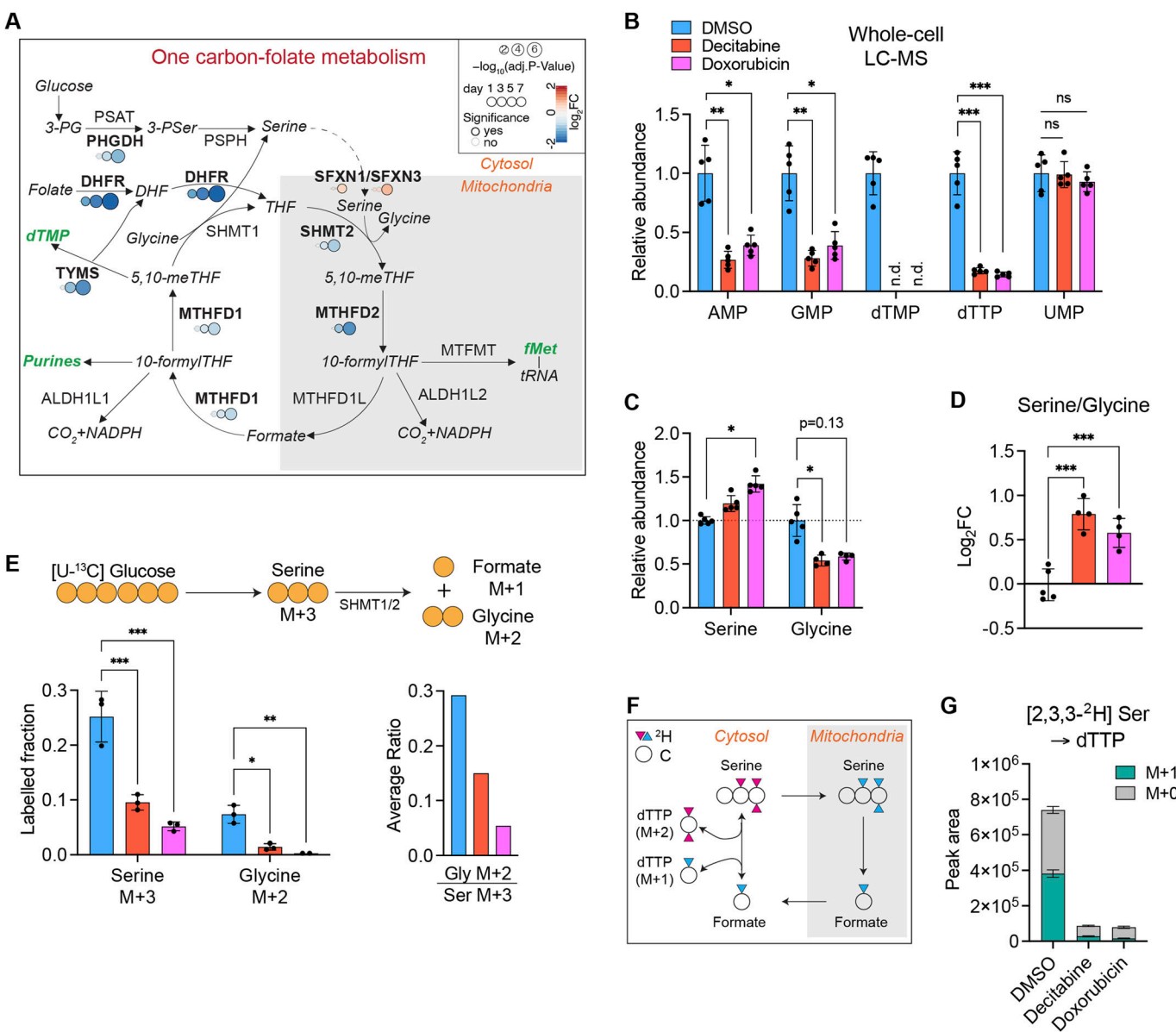

**Figure 6. Early down-regulation of 1C-folate metabolism coordinated between mitochondria and cytosol in senescent fibroblasts.**
**(A)** The mammalian 1C-folate metabolism. Enzyme levels are derived from the proteomics data. DEGs are shown in bold font. The color code indicates $Log_2$ fold changes. The major products of the pathway are shown in green italicized font. fMet, formyl-methionine. **(B)** Steady-state levels of nucleotides relevant to 1C-folate metabolism are shown. The statistical analysis was performed for $Log_2$ fold changes of each treatment compared with DMSO. See the Materials and Methods section and Fig S5. n.d.: not detected. Mean ± SD. Welch $t$ test, Bonferroni–Dunn correction. n = 5 from independent cultures. **(C, D)** Steady-state levels of serine and glycine and their ratio. One outlier in the measurement of glycine was excluded from both decitabine- and doxorubicin-treated samples. Mean ± SD. One-way ANOVA, Dunnett correction. n = 4–5 from independent cultures. **(E)** Metabolic tracing of serine and glycine using 5.5 mM [U-$^{13}$C] glucose added to the MEM ($^{13}$C:$^{12}$C = 1:1) for 24 h in IMR90 fibroblasts on day 7 after the treatment of DMSO, decitabine or doxorubicin. Mean ± SD. One-way ANOVA, Dunnett correction. n = 3 from independent cultures. **(F)** Schematic diagram of the serine metabolism highlighting the compartmentalized flux of carbons and hydrogens through the 1C-folate cycle. **(G)** Metabolic tracing of dTTP using [2,3,3-$^2$H$_3$] serine (200 $\mu$M) for 24 h in IMR90 fibroblasts on day 7 after the treatment with DMSO, decitabine or doxorubicin. Mean ± SD. One-way ANOVA, Dunnett correction. n = 3 from independent cultures.

when assessing mitochondrial activities. Our results revealed that senescent cells indeed accumulate mitochondria, although their bioenergetic activity is reduced. Measuring mitochondrial abundance in senescent cells, often employing two-dimensional images or $\Delta\Psi_m$-dependent fluorescent probes (11, 18, 19), may systematically underestimate the accumulation of mitochondria and explain conflicting observations on mitochondrial functions and fitness in these cells.

Together, our findings establish extensive reprogramming of the mitochondrial proteome and metabolism in senescent fibroblasts. We demonstrate increased BCAA degradation and lipid metabolism and decreased 1C-folate metabolism and OXPHOS activities associated with reduced mitochondrial translation in senescent cells. Because the mitochondria dictate the profile of the SASP, it is conceivable that metabolic rewiring of mitochondria is required for

and shapes the SASP and therefore may impact the effects of senescent cells in the context of age-related diseases such as cancer.

# Materials and Methods

### Cell culture and chemicals

Human lung IMR90 fibroblasts were obtained from ATCC (CCL-186) and maintained in minimum essential medium (MEM+glutaMAX, 41090; Thermo Fisher Scientific) supplemented with 9.5% FBS (F7524; Sigma-Aldrich). IMR90 cells were cultured under 3% $O_2$, 5% $CO_2$, and 92% $N_2$. Cells with SA-$\beta$-Gal positivity less than 10% of the population were used in all experiments. Upon the induction of CS, the medium was replaced every other day to exclude nutrient availability as a limiting factor for CS. For lentivirus production, HEK293T cells were maintained in DMEM (61965; Thermo Fisher Scientific) supplemented with 9.5% FBS. All cells were cultured without antibiotics and routinely checked for mycoplasma contamination. The cell number was calculated with trypan blue using Countess automated cell counter (Thermo Fisher Scientific). The chemicals used in the cell culture experiments are as follows: DMSO (D2650; Sigma-Aldrich), decitabine (ab120842; Abcam), doxorubicin (D1515; Sigma-Aldrich). Decitabine and doxorubicin were dissolved in DMSO and $H_2O$, respectively.

### Establishment of cellular senescence

IMR90 fibroblasts were seeded on a diverse size of culture vessels with a density of 2,100/$cm^2$ for DMSO (0.01% vol/vol) and decitabine (1 $\mu$M), or 6,500/$cm^2$ for doxorubicin (300 nM) treatment. Cells were treated with the compounds on the next day and, subsequently, the medium was replaced every other day. DMSO and decitabine were present in the media at all times, whereas doxorubicin was washed out after the first medium change. Unless denoted otherwise, the timing of cell harvest was synchronized to be 24 (±3) hours from the last medium replacement and DMSO-treated cells were timely re-plated so that they did not reach the confluence by the time of harvest to maintain a proliferating state. All senescence assays were performed 7 d after the initial treatment unless otherwise specified.

### Cell proliferation assay

Cells were incubated with 10 $\mu$M EdU in DMSO for 24 h corresponding to the population doubling time. Cells were collected by trypsinization and then processed according to the manufacturer's protocol (C10634; Thermo Fisher Scientific). The number of EdU-positive cells was counted by flow cytometry (FACS Canto; BD Biosciences) in the APC channel using conventional FSC/SSC gating criteria without a viability dye.

### Measurement of MMP, superoxide and polarized mitochondria by flow cytometry

Cells were seeded on a six-well plate with the density described above. On day 7, cells were collected by trypsinization and pelleted and then processed according to the manufacturer's protocol for labeling with mitoSOX (M36008; Thermo Fisher Scientific), TMRM (M20036; Thermo Fisher Scientific), and Mitotracker Deep Red FM (M22426; Thermo Fisher Scientific). Briefly, the collected cell pellets were resuspended in the 1 ml PBS with mitoSOX (5 $\mu$M), TMRM (20 nM) or Mitotracker Deep Red FM (50 nM) and incubated in a non-$CO_2$ incubator at 37°C for 20 min. Cells were pelleted and washed with PBS twice and DAPI (1 $\mu$g/ml) was added to select live cells. Then, cells were filtered through a 50-$\mu$m cell strainer and analyzed by flow cytometry (FACScanto; BD Biosciences) in the corresponding channels (PE or APC) with the conventional SSC/FSC single-cell gating strategy. The mean fluorescence intensity of the gated population was taken.

### Senescence-associated $\beta$-galactosidase assay

Cells were washed twice with PBS and subject to SA-$\beta$-Gal assay according to the manufacturer's protocol (ab65351; Abcam). On the next day, the cells were washed twice with PBS and permeabilized with 0.2% TX-100/PBS for 5 min. After washing twice, DAPI (1 ng/ml) was added to allow cell counting. The images were taken under the DAPI channel and transparent channel using an EVOS microscope (Thermo Fisher Scientific). At least 50 cells per condition were analyzed.

### Gene silencing by RNA interference

500,000 cells were seeded on 10-cm dishes, and simultaneously, esiRNA was reversely transfected using Lipofenctamine RNAiMAX (13778150; Thermo Fisher Scientific) and OPTI-MEM (51985034; Thermo Fisher Scientific) as per the manufacturer's protocol. Briefly, 9 $\mu$l of RNAiMAX was mixed with 300 $\mu$l OPTI-MEM. 10 min later, 1 $\mu$g esiRNA targeting either FLUC (control, EHUFLUC; Sigma-Aldrich) or BCKDHA (EHU230591; Sigma-Aldrich) was then added to the mixture and gently tapped. 15 min later, the mixture was added to the seeded cells in a drop-wise manner.

### Cell viability assay

Dead cells in the media and live cells were collected by trypsinization and pelleted by 2,500$g$ for 2 min. Cells were washed once with PBS, stained with DAPI (1 $\mu$g/ml), and subjected to flow cytometry analysis. After conventional FSC/SSC gating, the dead cells were gated as a DAPI-positive population.

### Quantification of mitochondrial volume

Cells were seeded and senescence was induced. DMSO-treated control cells were seeded the day before the assay was performed. On day 7, cells were washed twice with PBS and fixed with 4% PFA (sc-281692; Santa Cruz) for 15 min at RT. After washing out PFA with PBS twice, cells were permeabilized with 0.2% TX-100 for 5 min at RT. The cells were washed twice and incubated with the antibody against ATP5B (A21351; diluted 1:1,000 in 1% BSA/PBS; Invitrogen) overnight at 4°C. On the next day, the primary antibody was washed out and goat anti-mouse IgG (H+L) antibody conjugated with Alexa fluor 568 (A11031; Invitrogen) was added (1:1,000 in PBS with Alexa Fluor 647 Phalloidin [A22287; Invitrogen]) for F-actin staining to

identify single cells. After 1 h, DAPI (1 ng/ml) was added after washing out the secondary antibodies for 5 min and mounted on the slides (P10144; Thermo Fisher Scientific). At least one day after the mounting, the images were taken using a confocal microscope (SP8-DLS; Leica). Z-stack confocal images were taken with 0.2 $\mu m$ intervals from the bottom to the top of the mitochondria. After a single cell was defined in each image based on the F-actin staining using the software Fiji ([51]), the stacks of two-dimensional mitochondrial images were converted into the three-dimensional model by Mitograph 3.0 ([30]). The total length, average width, and volume (by length) of mitochondria per cell were calculated by MitoGraph 3.0.

### Real-time quantitative PCR (qRT–PCR)

RNA was harvested from the cells (740955; Macherey-Nagel) and subjected to cDNA synthesis with oligo(dT) reverse transcriptase (A2791; Promega) according to the manufacturer's protocol. Target mRNA levels were quantified by ΔΔCt values using TaqMan Fast Advanced Master Mix (4444557; Thermo Fisher Scientific) with the TaqMan probes as follows: B2M (Hs99999907_m1), IL1A (Hs00174092_m1), IL1B (Hs01555410_m1), IL6 (Hs00174131_m1), CDKN1A (Hs00355782_m1), CDKN2A (Hs00923894_m1), LMNB1 (Hs01059210_m1). Fold changes were calculated using B2M as a reference control.

### Western blot

Cells were scraped and lysed in RIPA buffer (20 mM Tris–HCl [pH 7.5], 150 mM NaCl, 1% NP-40, 1% sodium deoxycholate, 0.1% SDS), supplemented with protease and phosphatase inhibitor cocktail (78429 & 78420; Thermo Fisher Scientific). The lysate was cleared out by centrifugation at 20,000$g$ for 20 min at 4°C. Protein concentration from the supernatant was measured by BCA protein assay (23227; Thermo Fisher Scientific) and 20–30 $\mu g$ of protein was resolved in 12% Bis-Tris SDS–PAGE. The resolved protein was then transferred to the PVDF membrane and blocked with 5% skim milk diluted in TBS-T (20 mM Tris–HCL pH 7.4, 150 mM NaCl, 0.1% Tween 20). The membrane was then incubated with primary antibodies diluted in 5% skim milk TBS-T overnight at 4°C. The membrane was washed three times with TBS-T and incubated with goat anti-mouse or anti-rabbit IgG HRP-conjugated secondary antibody (1:10,000 in 5% BSA TBS-T, #1706515, #1706516; Bio-Rad) for 1 h, at RT. The membranes were washed again and developed using enhanced chemiluminescence and analyzed by ChemoStar Touch (INTAS Science Imaging). The primary antibodies used in the study are as follows: phospho-Rb$^{S807/811}$ (1:1,000, 8516; CST), Rb (1:000, 9309; CST), phospho-H2A.X$^{S139}$ (1:1,000, 9718; CST), LMNB1 (1:1,000, ab16048; Abcam), LMNA/C (1:2,000, 612162; BD Bioscience), CCNA2 (1:2,000, C4710; Sigma-Aldrich), p53 (1:1,000, 48818; CST), VCL (1:5,000, 4650; CST).

### Quantification of mtDNA copy number difference

Cellular DNA was extracted from the cells (69504; QIAGEN) and the mtDNA copy number was measured using the TaqMan assay as described above. Genomic DNA was measured using ACTB as a probe (Hs03023880_g1) and mtDNA was measured by two different probes (MT-ND1; Hs02596873_s1 and MT-7s; Hs02596861_s1). mtDNA copy number differences were calculated (MT-ND1/ACTB or MT-7s/ACTB) and represented by MT-ND1/ACTB as both values were comparable.

### Measurement of OCR

Mitochondrial respiration was measured using an XFe96 Seahorse analyzer (103015; Agilent) according to the manufacturer's protocol. Briefly, $2 \times 10^4$ (proliferating) and $3 \times 10^4$ (senescent) cells were seeded per well on XFe96 plate. The next day, cells were washed twice and incubated for 1 h at 37°C in the non-CO$_2$ chamber with 180 $\mu l$ of the assay medium (103575; Agilent) supplemented with L-glutamine (2 mM) and D-glucose (5.5 mM). OCR was measured with subsequent injections of the following compounds (1 $\mu M$ oligomycin, 0.5 $\mu M$ FCCP or CCCP, and rotenone and antimycin A [0.5 $\mu M$ each]). After the assay, the cells were washed once with PBS and lysed in 25 $\mu l$ of SDS buffer (50 mM Tris–HCl pH 7.4, 1% SDS), followed by the BCA protein quantification. OD$_{562nm}$ value was used as the protein amount without standards. The data were first normalized to protein amounts, followed by scaling to the cell number by a cell-to-protein ratio (Fig S1G). Spare respiratory capacity and proton leak were calculated from the OCR data by the Seahorse XF report generator (Agilent).

### Measurement of the cell-to-protein ratio

On day 7 after the treatment with H$_2$O, DMSO, decitabine or doxorubicin, IMR90 fibroblasts were trypsinized and the number of live cells was counted with trypan blue. The cells were pelleted and lysed in the SDS buffer (50 mM Tris–HCl pH 7.4, 1% SDS) and the protein mass was measured by the BCA method.

### Isolation of mitochondria

The preparation of mitochondria-enriched membrane organelle was done as described with a few modifications ([52]). Briefly, cells with around 80% confluence on three 15-cm dishes were collected by scraping and washed with ice-cold PBS twice. All subsequent steps were performed at 4°C. The cells were incubated for 10 min in 1 ml isolation buffer (10 mM HEPES-KOH pH 7.4, 225 mM mannitol, 75 mM sucrose, 1 mM EGTA). Then, cells were homogenized by passing 10 times through a 27G needle. The homogenates were spun down at 800$g$ for 5 min to remove cell debris. Supernatants were centrifuged at 7,000$g$ for 10 min, followed by two washing steps with an isolation buffer. The final pellets containing membrane organelles without cytosolic fraction were resuspended in the 200 $\mu l$ isolation buffer.

### Mitochondrial translation in organello assay

Equal amounts of mitochondria (100–150 $\mu g$) were resuspended in 1 ml translation buffer (60 $\mu g$/ml of each of 19 proteogenic amino acids except methionine, 5 mM ATP, 200 $\mu M$ GTP, 6 mM creatine phosphate, 60 $\mu g$/ml creatine kinase, 100 mM D-mannitol, 10 mM sodium succinate dibasic hexahydrate, 80 mM KCl, 5 mM MgCl$_2$ hexahydrate, 1 mM KH$_2$PO$_4$, 25 mM HEPES, adjusted to pH 7.4 with

KOH). 17 μl of $^{35}$S-methionine (SRM-01; Hartmann Analytic) was added and the sample was incubated for 1 h at 37°C under gentle mixing (300 rpm). Mitochondria were pelleted at 7,000$g$ for 2 min at 4°C and resuspended in translation buffer, followed by 10 min incubation at 37°C under gentle mixing (300 rpm). Mitochondria were washed three times with translation buffer to remove any residual $^{35}$S-methionine and then resuspended in 100 μl sample buffer and run on 12% Tris-tricine SDS–PAGE. The gel was transferred to a nitrocellulose membrane and dried in the air. The radioactivity was captured by the storage phosphor screen overnight and detected by the Typhoon Phosphorimager (Cytiva Lifesciences). To detect the protein amount in each sample, the blots were developed as described in the Western blot method with a slight modification. Briefly, membranes were blocked in 5% skim milk in TBS-T (20 mM Tris–HCL pH 7.4, 150 mM NaCl, 0.1% Tween 20) for 1 h, washed three times with TBS-T, and incubated with primary antibodies in 5% BSA TBS-T against SDHA (1:10,000, ab14715; Abcam), MT-CO1 (1:10,000, ab14705; Abcam), MT-ND1 (1:2,000, ab181848; Abcam), MT-CO2 (1:1,000, A6404; Invitrogen), MT-ATP8 (1:2,000, 26723-1-AP; Proteintech), MT-ATP6 (1:2,000, 55313-1-AP; Proteintech). Then, the membranes were washed three times with TBS-T and incubated with goat anti-mouse or anti-rabbit IgG HRP-conjugated secondary antibody (1:10,000 in 5% BSA TBS-T, #1706515, #1706516; Bio-Rad) for 1 h, at RT. The membranes were washed again and developed using enhanced chemiluminescence and analyzed by ChemoStar Touch (INTAS Science Imaging) and the Fiji software (51).

## Proteomics: peptide preparation

Cells were seeded on 15-cm dishes and, on the next day, treated with either DMSO or decitabine. On days 1, 3, 5, and 7, cells were collected and washed twice with PBS. In all conditions, the cells were collected at a confluency of around 80%. Cell pellets were resuspended in 15 μl of lysis buffer (6 M guanidinium chloride, 2.5 mM tris(2-carboxyethyl) phosphine, 10 mM chloroacetamide, 100 mM tris-hydrochloride) and heated at 95°C for 10 min. The lysates were sonicated (30 s/30 s, 10 cycles, high performance) by Bioruptor (B01020001; Diagenode), followed by centrifugation at 21,000$g$ for 20 min at 20°C. 200 μg of supernatants were digested with 1 μl trypsin (V5280; Promega) overnight at 37°C. On the next day, formic acid was added to the digested peptide lysates (to 1% final concentration) to stop trypsin digestion and samples were desalted by homemade STAGE tips (53). Eluted lysates in 60% acetonitrile/0.1% formic acid were dried by vacuum centrifugation (Concentrator Plus; Eppendorf) at 45°C.

## Proteomics: TMT labeling

4 μg of desalted peptides were labeled with tandem mass tags TMT10plex (90110; Thermo Fisher Scientific) using a 1:20 ratio of peptides to TMT reagent. TMT labeling was carried out according to the manufacturer's instruction with the following changes: dried peptides were reconstituted in 9 μl 0.1 M TEAB, to which, 7 μl TMT reagent in acetonitrile was added to a final acetonitrile concentration of 43.75%. The reaction was quenched with 2 μl 5% hydroxylamine. Labeled peptides were pooled, dried, resuspended in

0.1% formic acid, split into two samples, and desalted using homemade STAGE tips (53).

## Proteomics: high-pH fractionation

Pooled TMT-labeled peptides were separated on a 150 mm, 300 μm OD, 2 μm C18, Acclaim PepMap (Thermo Fisher Scientific) column using an Ultimate 3000 (Thermo Fisher Scientific). The column was maintained at 30°C. Separation was performed with a flow of 4 μl using a segmented gradient of buffer B from 1–50% for 85 min and 50–95% for 20 min. Buffer A was 5% acetonitrile 0.01 M ammonium bicarbonate, buffer B was 80% acetonitrile 0.01 M ammonium bicarbonate. Fractions were collected every 150 s and combined into nine fractions by pooling every ninth fraction. Pooled fractions were dried in Concentrator plus (Eppendorf), and resuspended in 5 μl 0.1% formic acid, from which 2 μl were analyzed by LC-MS/MS.

## Proteomics: LC-MS/MS analysis

Dried fractions were resuspended in 0.1% formic acid and separated on a 50 cm, 75 μm Acclaim PepMap column (164942; Thermo Fisher Scientific) and analyzed on an Orbitrap Lumos Tribrid mass spectrometer (Thermo Fisher Scientific) equipped with a FAIMS device (Thermo Fisher Scientific). The FAIMS device was operated in two compensation voltages, –50 and –70 V. Synchronous precursor selection based on MS3 was used for the acquisition of the TMT reporter ion signals. Peptide separation was performed on an EASY-nLC1200 using a 90 min linear gradient from 6–31% buffer; buffer A was 0.1% formic acid, and buffer B was 0.1% formic acid with 80% acetonitrile. The analytical column was operated at 50°C. Raw files were split based on the FAIMS compensation voltage using FreeStyle (Thermo Fisher Scientific).

## Proteomics: peptide identification and quantification

Proteomics data were analyzed using MaxQuant, version 1.5.2.8 (54). The isotope purity correction factors, provided by the manufacturer, were included in the analysis. Mitochondrial annotations were based on human MitoCarta 3.0 (35).

## Proteomics: data analysis and visualization

The proportion of proteins, corresponding to a certain subcellular compartment, in the total proteome was calculated by dividing the sum of the TMT reporter intensities for all proteins which correspond to that compartment by the sum of the TMT reporter intensities for all quantified proteins. Before differential expression analysis, TMT reporter intensities were normalized within a TMT multiplex using VSN (55). Intensity normalization and differential expression analysis were carried out using proteins quantified in all 32 samples, total peptide counts, or using the subset of mitochondrial proteins only, mitochondrial-specific peptide counts. Thus, the fold change denotes the change in protein abundance within a given (sub-)proteome. Differential expression analysis was performed using limma version 3.34.9 (56) and R version 3.4.3 (57). Proteins with $P < 0.05$ (Bonferroni–Hochberg method) were deemed significant and differentially expressed. Quantified proteomics data

were investigated for the enrichment analysis including statistics by the String database ([58]) and the GSEA ([59], [60]). For the GSEA analysis, the background gmt files were made with MitoPathways and localization information from the human MitoCarta 3.0. The total quantified 6,482 proteins were used as a background. For the categorization of the organellar proteome, the reference proteome was used from the publicly available data as described in the main text ([Fig 3B]). Graphs were drawn by GraphPad Prism version 9.3.1 and [Figs S3] and [S4] by R version 3.4.3.

## Metabolomics: metabolite preparation

Cells on six-well plates were washed twice with the wash buffer (75 mM ammonium carbonate, pH 7.4) and the plates were flash-frozen in liquid nitrogen. 800 $\mu$l extraction buffer (acetonitrile:methanol:$H_2O$ = 4:4:2, −20°C) was added to the wells, scraped, and centrifuged by 21,000$g$ for 20 min at 4°C. The supernatants were dried by vacuum centrifugation (Labogene) for 6 h at 20°C. Pellets were lysed in 50 mM Tris-KOH pH 8.0, 150 mM NaCl, 1% SDS, and used for protein quantification using the BCA assay (23225; Thermo Fisher Scientific). To measure steady-state levels of metabolites, the following internal standards were added to the extraction buffer: 2.5 mM amino acid standard (MSK-A2-1.2; CIL), 100 $\mu$g/ml citrate d$_4$ (485438; Sigma-Aldrich), 1 mg/ml $^{13}C_{10}$ ATP (710695; Sigma-Aldrich). No internal standard was added for the stable isotope-tracing experiments. Isotopologues used in the experiments are as follows: $^{13}C_6$ D-glucose (389374; Sigma-Aldrich), 2,3,3-$^2H_3$ L-serine (DLM-582; CIL), $^{13}C_6$ L-leucine (605239; Sigma-Aldrich), $^{15}N$ L-leucine (340960; sigma-Aldrich), $^{13}C_5$ L-valine (758159; Sigma-Aldrich), $^{15}N$ L-valine (490172; Sigma-Aldrich). Isotopologues were added to the regular culture medium (MEM supplemented with 9.5% undialyzed FBS) and treated to cells as indicated in each figure legend.

## Metabolomics: anion-exchange chromatography mass spectrometry for the analysis of anionic metabolites

Extracted metabolites were resuspended in 200 $\mu$l of Optima UPLC/MS grade water (Thermo Fisher Scientific). After 15 min incubation on a thermomixer at 4°C and a 5 min centrifugation at 16,000$g$ at 4°C, 100 $\mu$l of the cleared supernatant was transferred to polypropylene autosampler vials (Chromatography Accessories Trott). The samples were analyzed using a Dionex ion chromatography system (Integrion, Thermo Fisher Scientific) as described previously ([61]). In brief, 5 $\mu$l of the polar metabolite extract were injected in full-loop mode using an overfill factor of 1, onto a Dionex IonPac AS11-HC column (2 mm × 250 mm, 4 $\mu$m particle size, Thermo Fisher Scientific) equipped with a Dionex IonPac AG11-HC guard column (2 mm × 50 mm, 4 $\mu$m, Thermo Fisher Scientific). The column temperature was held at 30°C, whereas the autosampler was set to 6°C. A potassium hydroxide gradient was generated using a potassium hydroxide cartridge (Eluent Generator, Thermo Fisher Scientific), which was supplied with deionized water. The metabolite separation was carried out at a flow rate of 380 $\mu$l/min, applying the following gradient conditions: 0–3 min, 10 mM KOH; 3–12 min, 10–50 mM KOH; 12–19 min, 50–100 mM KOH, 19–21 min, 100 mM KOH, 21–22 min, 100-10 mM KOH. The column was re-equilibrated at 10 mM for 8 min.

For the analysis of metabolites and their isotopologues, the eluting compounds were detected in negative ion mode using full scan measurements in the mass range m/z 50–750 on a Q-Exactive HF high-resolution MS (Thermo Fisher Scientific). The heated electrospray ionization (ESI) source settings of the mass spectrometer were: spray voltage 3.2 kV, the capillary temperature was set to 275°C, sheath gas flow 50 AU, aux gas flow 14 AU at a temperature of 380°C, and a sweep gas flow of 3 AU. The S-lens was set to a value of 45.

The semi-targeted LC-MS data analysis was performed using the TraceFinder software (Version 4.1; Thermo Fisher Scientific). The identity of each compound was validated by authentic reference compounds, which were measured at the beginning and the end of the sequence.

For data analysis, the area of the deprotonated [M-H+]− monoisotopic mass peak and the corresponding isotopologues of each compound were extracted and integrated using a mass accuracy of <5 ppm and a retention time (RT) tolerance of <0.05 min as compared with the independently measured reference compounds. Areas of the cellular pool sizes were normalized to the internal standards added to the extraction buffer, followed by total ion count (TIC) normalization.

## Metabolomics: semi-targeted liquid chromatography-high-resolution mass spectrometry-based (LC-HRS-MS) analysis of amine-containing metabolites

The LC-HRMS analysis of amine-containing compounds was performed using an adapted benzoylchloride-based derivatization method ([62]). In brief, the polar fraction of the metabolite extract was resuspended in 200 $\mu$l of LC-MS-grade water (Optima-Grade; Thermo Fisher Scientific) and incubated at 4°C for 15 min on a thermomixer. The resuspended extract was centrifuged for 5 min at 16,000$g$ at 4°C and 50 $\mu$l of the cleared supernatant was mixed with 25 $\mu$l of 100 mM sodium carbonate (Sigma-Aldrich), followed by the addition of 25 $\mu$l 2% (vol/vol) benzoylchloride (Sigma-Aldrich) in acetonitrile (Optima-Grade; Thermo Fisher Scientific). Samples were vortexed and kept at 20°C until analysis. For the LC-HRMS analysis, 1 $\mu$l of the derivatized sample was injected onto a 100 × 2.1 mm HSS T3 UPLC column (Waters). The flow rate was set to 400 $\mu$l/min using a binary buffer system consisting of buffer A (10 mM ammonium formate [Sigma-Aldrich]), 0.15% (vol/vol) formic acid (Sigma-Aldrich) in LC-MS-grade water (Optima-Grade; Thermo Fisher Scientific) and buffer B. Buffer B consisted solely of acetonitrile (Optima-grade; Thermo Fisher Scientific). The column temperature was set to 40°C, whereas the LC gradient was: 0% B at 0 min, 0–15% B 0–4.1 min; 15–17% B 4.1–4.5 min; 17–55% B 4.5–11 min, 55–70% B 11–11.5 min, 70–100% B 11.5–13 min; B 100% 13–14 min; 100-0% B 14–14.1 min; 0% B 14.1–19 min; 0% B. The mass spectrometer (Q-Exactive Plus; Thermo Fisher Scientific) was operating in positive ionization mode recording the mass range m/z 100–1,000. The heated ESI source settings of the mass spectrometer were: spray voltage 3.5 kV, capillary temperature 300°C, sheath gas flow 60 AU, aux gas flow 20 AU at a temperature of 330°C, and the sweep gas to 2 AU. The RF lens was set to a value of 60. Semi-targeted data analysis for the samples was performed using the TraceFinder software (Version 4.1; Thermo Fisher

Scientific). The identity of each compound was validated by authentic reference compounds, which were run before and after every sequence. Peak areas of [M+nBz+H]+ ions were extracted using a mass accuracy (<5 ppm) and a retention time tolerance of <0.05 min. Areas of the cellular pool sizes were normalized to the internal standards ([U]-$^{15}$N;[U]-$^{13}$C amino acid mix [MSK-A2-1.2]; Cambridge Isotope Laboratories), which were added to the extraction buffer, followed by normalization to the TIC.

### Metabolomics: semi-targeted liquid chromatography-high-resolution mass spectrometry-based (LC-HRS-MS) analysis of Acyl-CoA metabolites

The LC-HRMS analysis of Acyl-CoAs was performed using a modified protocol based on the previous method (62). In brief, the polar fraction of the metabolite extract was resuspended in 50 $\mu$l of LC-MS-grade water (Optima-Grade; Thermo Fisher Scientific). For the LC-HRMS analysis, 1 $\mu$l of the sample was injected onto a 30 × 2.1 mm BEH Amide UPLC column (Waters) with a 1.7-$\mu$m particle size. The flow rate was set to 500 $\mu$l/min using a quaternary buffer system consisting of buffer A (5 mM ammonium acetate; Sigma-Aldrich) in LC-MS-grade water (Optima-Grade; Thermo Fisher Scientific). Buffer B consisted of 5 mM ammonium acetate (Sigma-Aldrich) in 95% acetonitrile (Optima-grade; Thermo Fisher Scientific). Buffer C consisted of 0.1% phosphoric acid (85%; VWR) in 60% acetonitrile (acidic wash) and buffer D of 50% acetonitrile (neutral wash). The column temperature was set to 30°C, whereas the LC gradient was: 85% B for 1 min, 85–70% B 1–3 min; 70–50% B 3–3.2 min; holding 50% B till 5 min; 100% C 5.1–8 min, 100% D 8.1–10 min; followed by re-equilibration 85% B 10.1–13 min. The mass spectrometer (Q-Exactive Plus; Thermo Fisher Scientific) was operating in positive ionization mode recording the mass range m/z 760–1800. The heated ESI source settings of the mass spectrometer were: spray voltage 3.5 kV, capillary temperature 300°C, sheath gas flow 50 AU, aux gas flow 15 AU at a temperature of 350°C, and the sweep gas to 3 AU. The RF lens was set to a value of 55.

Semi-targeted data analysis for the samples was performed using the TraceFinder software (Version 4.1; Thermo Fisher Scientific). The identity of Acetyl-CoA and Malonyl-CoA was validated by authentic $^{13}$C-labelled reference compounds, which were run before. Other Acyl-CoAs were validated by using *E. coli* reference material matching exact mass and reporter ions from PRM experiments. Peak areas of [M+H]+ ions and corresponding isotopomers were extracted using a mass accuracy (<5 ppm) and a retention time tolerance of <0.05 min. The peak area was normalized by the TIC.

### Metabolomics: data analysis and visualization

The steady-state level of metabolites was normalized by the TIC value. Statistical analysis of differential abundance was performed with fold changes in log$_2$ values by the Welch $t$ test with correction using the Bonferroni–Dunn method. For mass isotopologue experiments, the natural abundance of $^{13}$C was not corrected and the kinetic isotope effect of the $^2$H tracer was not considered. All the statistical analyses and graphs were done by GraphPad Prism version 9.3.1. For the heatmap in Fig S5C, Flaski was used (63).

### Data analysis and statistics

All statistical analyses were performed by GraphPad Prism version 9.3.1 except proteomics data. When two groups were compared, the Welch $t$ test was used with a multiple comparison correction by the Bonferroni–Dunn method, if needed. When three or more groups were compared, the ordinary ANOVA test was used. One-way ANOVA was used for multiple groups under one condition and two-way ANOVA for multiple groups under two conditions. Each subject group was compared with the control group with a multiple comparison correction by the Dunnett method. *$P$ < 0.05, **$P$ < 0.01, ***$P$ < 0.001.

# Supplementary Information

# Acknowledgements

Foremost, we would like to thank Dr. Patrick Giavalisco in the metabolomics core facility at Max Planck Institute for Biology of Ageing for having given critical advice and fostering insightful discussion throughout the study. We would also like to thank Kat Folz-Donahue and other staff in the FACS & Imaging core facility at Max Planck Institute for Biology of Ageing for the technical assistance. We thank Dr. Jorge Boucas, Ayesha Iqbal, and Franziska Metge in the bioinformatics core facility at Max Planck Institute for Biology of Ageing for their help to analyze and visualize the omics data. Lastly, we appreciate the critical discussion about the study from Alvaro J Narbona Perez, Dr. Srikanth Chandragiri, Nils Grotehans, Dr. Amir Bahat, and other members of Thomas Langer's laboratory.

### Author Contributions

JY Kim: conceptualization, data curation, formal analysis, validation, investigation, visualization, project administration, and writing—original draft, review, and editing.
I Atanassov: data curation, software, formal analysis, visualization, and methodology.
F Dethloff: data curation, formal analysis, and methodology.
L Kroczek: validation and methodology.
T Langer: conceptualization, supervision, funding acquisition, project administration, and writing—original draft, review, and editing.

### Conflict of Interest Statement

The authors declare that they have no conflict of interest.

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
