## [Reviewer comments · Life Science Alliance]

Life Science Alliance

Time-resolved proteomic analyses of senescence highlights rewiring of mitochondrial metabolism

Jun Yong Kim, Ilian Atanassov, Frederik Dethloff, Lara Kroczek, and Thomas Langer

DOI: <https://doi.org/10.26508/lsa.202302127>

Corresponding author(s): Thomas Langer, Max Planck Institute for Biology of Ageing and Jun Yong Kim, Max Planck Institute for Biology of Ageing

Review Timeline:

Submission Date:	2023-05-02
Editorial Decision:	2023-05-31
Revision Received:	2023-06-07
Accepted:	2023-06-08

Transaction Report:

Please note that the manuscript was reviewed at Review Commons and these reports were taken into account in the decision-making process at *Life Science Alliance*.

Review
COMMONS

Review #1

In the manuscript "Metabolic rewiring of mitochondria in senescence revealed by the time resolved analysis of the mitochondrial proteome", Kim et al performed the proteomic approach to investigate the mitochondrial programming during senescence induction of cultural fibroblast. The main findings mentioned by author are the enhanced catabolism of branched-chain amino acids and the reduction of 1C-folate metabolism.

1. The author claimed their approaches are "time-resolved analysis" from day 1 to day 7 during the senescence induction. However, they did not adequately describe the transition of molecular signatures and cellular states from normal cells to senescent cells in each timepoint. The authors need to quantify the protein levels of p16, p21, and p53 and transcript levels of SASP factors in each timepoint to define the stages during induction and discuss how their finding related to this transition.
2. The authors treated decitabine and doxorubicin in different periods. Sometimes they properly prepared two different control groups, and sometimes they use only one control group. This issue should be unified.
3. The mitochondria dysfunction and the increasing mitochondrial mass in cellular senescence have been reported in cell culture study previously. The authors described that the mitochondrial activity normalized to mitochondria volume decreased in senescent cells, which raises the importance of the exact value (mitochondria volume) they used for normalization. However, the quantification result in Figure 1 shows obvious batch effects and variation which make their estimation not reliable enough. The authors should increase replicate numbers in image analysis to provide robust quantification results. Besides, the orthogonal method for measuring mitochondrial mass should be also performed to confirm that the FC of volume is reliable.
4. The normalization and scaling strategy of proteomic data were not described in method. Senescent cells show increasing cell size and protein abundance. The authors should describe how they processed the peptide counts in detail. It is confused whether the up regulation ($FC > 1$) represents the increment per cell, per mitochondrion, or in the protein abundance.
5. The protein amounts of CDKN1A and CDKN2A were not increased on day 7 in Sup Fig 2C. The authors need to provide the explanation of them to prevent bias in their cellular senescence-dependent findings. In addition, ARF and INK4a should be separately quantified.
6. This comment is also related to comment #4. In line 223, the authors mentioned "This is accompanied by a decrease in mitochondrial translation, consistent with the observed decreased respiratory activity of mitochondria in senescent cells.". First, the linkage between mitochondrial translation and respiratory activity should be further illustrated. Second, the results in Fig. 2G and 2H clearly showed that the overall respiratory activity was enhanced in senescent cells (day 7), and only the activity normalized by mitochondrial volume showed decreased. How the authors normalize proteomic data from whole cell lysates with mitochondrial mass is missed in this manuscript.

Overall, the central findings of the study provide informative evidence to support several concepts and biological observations in previous reports. However, because of the lack of sufficient description and experimental verification, part of the conclusion is not rigorous enough and needed to be further improved.

Review #2

Kim et al. applied a new method to demonstrate that senescent cells accumulate with dysfunctional mitochondria. They isolated mitochondrial to profile mitochondrial proteome in a time-dependent manner and revealed a metabolic shift in mitochondria during the senescence process. However, many groups have widely studied mitochondrial biology in senescent cells (Joao F. Passos and Thomas Von Zglinicki et al.). Based on the previous finding, cellular senescence is also known for dramatic changes in mitochondrial mass, dynamics, structure, metabolism, and function. Thus, the current work is descriptive and incremental. I have specific comments listed below:

****Major:****

1. The authors stated that mitochondrial DNA (mtDNA) was decreased per mitochondrion. However, in Fig 2A, there is no statistical significance. So, this statement is not valid.
2. The authors investigated the mitochondrial proteome alterations during CS development (days 1 to 7 post-treatment of decitabine). However, in supplemental Fig. 2C, p16 and p21 did not increase on day 7. So, the mitochondria authors studied are the authors studied in a pre-senescent state. What is the rationale for this study, and why did the authors not examine mitochondria at the fully senescent stage?

3. The authors claimed that the fraction of mitochondrial proteins did not significantly change (Shown in Fig. 3B). Again, this could be due to cells at the pre-senescent stage; I wonder if this change could be significant after cells are fully senescent.
4. The data in Fig. 4E is not statistically significant. Please increase n to confirm your conclusion.
5. Are these mitochondrial proteome alterations associated with the senescent cell's functional output or other features? Are these changes cell type-dependent?
6. I suggest the authors clarify explicitly the knowledge gap that the current study accomplished.

****Minor:**** Please carefully check the statistical analysis. N of 2 is not sufficient for One-way ANOVA.

I do not see the significance of the current work. The work is descriptive and incremental, which reduce the impact of this manuscript.

Review #3

The manuscript by Kim and colleagues investigated the metabolic and proteomic changes in senescent fibroblasts. The authors performed a time-resolved analysis of the proteome and revealed the impacts on mitochondrial proteome. They also applied tracing approaches to further demonstrate the impact on mitochondrial metabolism and revealed a potential impact of branched-chain amino acid catabolism and carbon-folate metabolism in senescent fibroblasts. They concluded that the reprogramming of mitochondria influences the senescence-associated secretory phenotype (SASP) impacting diseases associated with senescent cells.

1. The authors quantified the volume of mitochondria to examine mitochondrial functions. Some data were normalized "per cell" and "per mito". Should it read per mito volume? The comparison between the two normalization procedures in Fig 2 is important but also confusing. Can the authors speculate why non-mitochondrial respiration seems to be increased in decitabine/doxorubicin conditions (Fig. 2G/H)? Are the data significantly affected after normalization to non-mitochondrial respiration?
2. The tracing data are impactful and critical to confirm metabolic changes. Can the authors explain why they added the tracer to the regular growth medium (instead of substituting the metabolite of interest)? The media composition of the proteomic and the metabolic experiments is not identical. For instance, additional 5.5mM ¹³C glucose has been added to the ¹³C glucose tracer experiment while all other experiments were performed with 5.5mM ¹²C glucose only. Changes in media composition certainly affect cell function and metabolism. The authors may want to repeat key tracing experiments to mimic experimental conditions used in proteomics analysis.
3. The impact on branched-chain amino acids is interesting. Did the authors observe ¹³C incorporation into the TCA cycle from BCAA? Increased BCAA catabolism may increase mitochondrial respiration, but the authors observed decreased OCR in senescent cells. Further, does inhibition of BCAA catabolism rescue the phenotype observed in senescent cells?

Understanding the metabolic reprogramming of mitochondria in senescent cells is interesting and of high interest to the research community. However, some clarification is needed on experimental conditions, as media compositions in proteome and metabolome experiments were different which certainly affects cell metabolism and function

Corresponding author(s): Thomas, Langer

1. General Statements

We have performed a comprehensive and time-resolved analysis of the mitochondrial proteome upon induction of cellular senescence and observed rapid metabolic rewiring of mitochondria. In particular, we identified two metabolic pathways, the 1C-folate metabolism and the branched-chain amino acid (BCAA) catabolism that were rapidly rewired upon senescence induction. Our analysis therefore provides unprecedented insight into the function and metabolic adaptations of mitochondria in senescent cells, which could also serve as a reference for future studies. Moreover, we demonstrate that quantification of mitochondrial abundance using the mitochondrial volume allows to reconcile seemingly contradicting conclusions on the function/fitness of mitochondria in senescent cells. In our opinion, our studies therefore represent a valuable and significant contribution to our understanding of the role of mitochondria in senescence.

Reviewer #1 (Evidence, reproducibility and clarity (Required)):

In the manuscript "Metabolic rewiring of mitochondria in senescence revealed by the time resolved analysis of the mitochondrial proteome", Kim et al performed the proteomic approach to investigate the mitochondrial programming during senescence induction of cultural fibroblast. The main findings mentioned by author are the enhanced catabolism of branched-chain amino acids and the reduction of 1C-folate metabolism.

1. The author claimed their approaches are "time-resolved analysis" from day 1 to day 7 during the senescence induction. However, they did not adequately describe the transition of molecular signatures and cellular states from normal cells to senescent cells in each timepoint. The authors need to quantify the protein levels of p16, p21, and p53 and transcript levels of SASP factors in each timepoint to define the stages during induction and discuss how their finding related to this transition.

As per the reviewer's requests, we monitored the molecular markers during the transition to the senescent state by decitabine. The mRNA level of CDKN1A (encoding p21) was significantly increased on days 5 and 7, while that of CDKN2A (encoding p16) was moderately but significantly increased (~40 %) on day 7 (**updated Suppl. Figure 1B**).

Furthermore, mRNA levels of two core SASP genes, IL1A and IL6 were significantly increased on day 7.

These results complement our analysis of protein levels of senescent markers including p16 and p21 and several SASP proteins (**updated Suppl. Figure 2C**). Of note, decreased intracellular protein levels of HMGB1 and HMGB2 indicate a senescent rather than a pre-senescent state (PMID: 27700366; 23649808) and demonstrate that cells display transcriptomic signatures of a senescent state after decitabine treatment for seven days. However, the mass spectrometric analysis of the cellular proteome revealed an accumulation of p21 on day 5 but not on day 7 (**updated Suppl. Figure 2C**). We therefore performed additional experiments to further corroborate the senescent state of the cells under these conditions and to collectively address concerns raised by reviewer 1 (points 1, 5) and reviewer 2 (points 2, 3). Senescence is defined by an irreversible cell-cycle arrest. Consistently, cells maintained the proliferation-deficient state and did not re-enter the cell cycle for 7 days after the removal of decitabine (**updated Suppl. Figure 1D**). Moreover, cells reached a plateau of SA-b-Gal positivity on day 7, which was maintained after 7 days of decitabine removal (**updated Suppl. Figure 1E**). We also examined the protein levels of several senescence markers (**updated Suppl. Figure 1F**). DNA damage was acutely induced upon decitabine treatment as evidenced by phosphorylated H2A.X at serine 139 (p-H2A.X^{S139}). This was accompanied by the rapid reduction of lamin B1 (LMNB1) which is a hallmark of senescence (PMID: 22496421). In agreement with the EdU data (**updated Suppl. Figure 1D**), the loss of phosphorylated Rb at serine 807/811 (p-Rb^{S807/811}) and cyclin A2 (CCNA2), essential for G1-S transition, from day 3 indicated a cell cycle-arrested state.

Together, we conclude that cells start to become senescent on day 5 and reach the senescent state on day 7 after decitabine treatment. We now show these data in the **updated Supplementary figure 1** and modified the text accordingly.

2. The authors treated decitabine and doxorubicin in different periods. Sometimes they properly prepared two different control groups, and sometimes they use only one control group. This issue should be unified.

Proteomic analysis revealed that DMSO treatment for 7 days did not affect the cellular proteome (**Reviewer figure 1**), indicating that DMSO is as ineffective as H₂O in our experimental conditions. When cells treated with decitabine and doxorubicin were assayed in the same experiment, cells treated with DMSO were used as a common control. When decitabine and doxorubicin-treated cells were assayed independently, either DMSO or H₂O was used as the control accordingly.

Reviewer figure 1. The pairwise comparison of proteome from cells treated with DMSO at different time points. The x-axis denotes \log_2FC and y-axis does $-\log(\text{adj.P-value})$. The data are of the same origin from the updated Suppl. Figure 2.

3. The mitochondria dysfunction and the increasing mitochondrial mass in cellular senescence have been reported in cell culture study previously. The authors described that the mitochondrial activity normalized to mitochondria volume decreased in senescent cells, which raises the importance of the exact value (mitochondria volume) they used for normalization. However, the quantification result in Figure 1 shows obvious batch effects and variation which make their estimation not reliable enough. The authors should increase replicate numbers in image analysis to provide robust quantification results. Besides, the orthogonal method for measuring mitochondrial mass should be also performed to confirm that the FC of volume is reliable.

We have increased the number of experiments quantifying the mitochondrial volume in senescent cells (**updated Figure 1B**). Despite some variation between experiments, we observed a significant increase in mitochondrial volume, which could be explained by an increased length of mitochondrial tubules, while the mitochondrial width remained unaltered. New quantification revealed an about 8-fold increase of mitochondrial volume both upon decitabine and doxorubicin treatment (about 30% lower than our original calculation). These experiments are now shown as violin plots, showing the distribution of the experimental data along with the median and quartile values (**updated Figure 1B**). They further substantiate our conclusion that the bioenergetic activity of mitochondria is increased per senescent cell, but decreased when calculated relative to the mitochondrial volume.

4. The normalization and scaling strategy of proteomic data were not described in method. Senescent cells show increasing cell size and protein abundance. The authors should describe how they processed the peptide counts in detail. It is confused whether the up regulation ($FC > 1$) represents the increment per cell, per mitochondrion, or in the protein abundance.

We apologize for not being clear at this point. For clarification, we have amended the text in the method part on p. 22:

‘The proportion of proteins of a certain subcellular compartment with the total cellular proteome was calculated by dividing the sum of the TMT reporter intensities for all proteins of this compartment by the sum of TMT reporter intensities for all quantified proteins. Prior to differential expression analysis, TMT reporter intensities were normalized to within the TMT multiplex using VSN (PMID: 12169536). Intensity normalization and differential expression analysis was carried out using proteins quantified in all 32 samples (total peptide counts) or using the subset of mitochondrial proteins only (mitochondria-specific peptide counts). Thus, the fold-change denotes the change in protein abundance within a given (sub-)proteome.’

5. The protein amounts of CDKN1A and CDKN2A were not increased on day 7 in Sup Fig 2C. The authors need to provide the explanation of them to prevent bias in their cellular senescence-dependent findings. In addition, ARF and INK4a should be separately quantified.

We have performed a series of additional experiments to corroborate that cells are senescent on day 7 (see point 1). We have not detected p14^{ARF}, but p16^{INK4A} in our proteomic analysis. CDKN2A thus refers to p16^{INK4A}. We have updated **Supplementary figure 2C** for clarification.

6. This comment is also related to comment #4. In line 223, the authors mentioned "This is accompanied by a decrease in mitochondrial translation, consistent with the observed decreased respiratory activity of mitochondria in senescent cells.". First, the linkage between mitochondrial translation and respiratory activity should be further illustrated. Second, the results in Fig. 2G and 2H clearly showed that the overall respiratory activity was enhanced in senescent cells (day 7), and only the activity normalized by mitochondrial volume showed decreased. How the authors normalize proteomic data from whole cell lysates with mitochondrial mass is missed in this manuscript.

Mitochondrial DNA encodes essential subunits of OXPHOS complexes. Therefore, decreased mitochondrial translation is accompanied by reduced respiratory activity. We have clarified this point on p. 13 of the revised manuscript.

The proteomic data show the abundance of mitochondrial proteins within the cellular proteome. Our proteomic analysis established that the mitochondrial protein mass increases proportionally to that of the cell (**updated Figure 3B**), consistent with the increase in mitochondrial volume. For clarification, we have now improved the description of our quantification method in the manuscript on p. 22.

Reviewer #1 (Significance (Required)):

Overall, the central findings of the study provide informative evidence to support several concepts and biological observations in previous reports. However, because of the lack of sufficient description and experimental verification, part of the conclusion is not rigorous enough and needed to be further improved.

To address the concerns of the reviewer, we provide now more experimental evidence that the cells attain the senescent state when analyzed by time-resolved proteomics and improved the description of our experimental procedures.

Reviewer #2 (Evidence, reproducibility and clarity (Required)):

Kim et al. applied a new method to demonstrate that senescent cells accumulate with dysfunctional mitochondria. They isolated mitochondrial to profile mitochondrial proteome in a time-dependent manner and revealed a metabolic shift in mitochondria during the senescence process. However, many groups have widely studied mitochondrial biology in senescent cells (Joao F. Passos and Thomas Von Zglinicki et al.). Based on the previous finding, cellular senescence is also known for dramatic changes in mitochondrial mass, dynamics, structure, metabolism, and function. Thus, the current work is descriptive and incremental.

We kindly disagree with this statement of the reviewer. We do appreciate and acknowledge previous studies on mitochondria in senescent cells, but are convinced that our work provides novel insight, as to how mitochondria are affected upon establishment of senescence. Rather than considering mitochondria as dysfunctional, our proteomic and metabolomic data indicate reprogramming of mitochondria: reduced respiratory activities are accompanied by enhanced BCAA catabolism, fatty acid metabolism, and Ca²⁺ transport (**updated Figure 4A, updated Figure 5**).

We would also like to point out that several reports on the role of mitochondria in senescent cells were seemingly contradicting and difficult to reconcile with each other. Some studies observed a decreased membrane potential and increased ROS production in senescent cells, while at the same time, mitochondrial respiration appeared to be increased (PMID: 15018610, 20160708). Other studies reported an increased OXPHOS activity due to enhanced mitochondrial metabolism of glucose/pyruvate and fatty acids (PMID: 23945590, 23685455, 30778219, 22421146). Our work provides a rationale for these apparently disparate findings, highlighting that the assessment of mitochondrial function/fitness in senescent cells requires the accurate determination of mitochondrial mass/volume. We show that senescent cells harbor more mitochondria with reduced respiratory activity.

As a side note, we did not isolate mitochondria but determined the cellular proteome and analyzed organellar proteomes (such as the mitochondrial proteome) based on reference databases (see also **updated Suppl. Figure 2D**).

I have specific comments listed below:

Major:

1. The authors stated that mitochondrial DNA (mtDNA) was decreased per mitochondrion. However, in Fig 2A, there is no statistical significance. So, this statement is not valid.

We have determined mtDNA levels in two additional, independent experiments but did not observe a statistically significant decrease in senescent cells due to a relatively high experimental variability (n=5). We have updated **Figure 2A** and corrected the text accordingly on p. 6.

2. The authors investigated the mitochondrial proteome alterations during CS development (days 1 to 7 post-treatment of decitabine). However, in supplemental Fig. 2C, p16 and p21 did not increase on day 7. So, the mitochondria authors studied are the authors studied in a pre-senescent state. What is the rationale for this study, and why did the authors not examine mitochondria at the fully senescent stage?

We kindly disagree with the statement of the reviewer that analyzed was a pre-senescent state. As outlined in detail in reply to reviewer 1 (point 1), the analysis of a series of established senescent markers on mRNA and protein levels demonstrates that cells have reached the senescent state after seven days of drug treatment. Most importantly, we show by EdU staining in Suppl. Figure 1D that cells are in cell cycle arrest on day 7 and now provide additional evidence (EdU- and SA-b-Gal staining, senescence and cell cycle markers) that this state is maintained for another seven days, excluding a quiescent state (**updated Suppl. Figure 1D, 1E**). This includes the loss of lamin B1, which is associated with senescence but not quiescence (PMID: 22496421).

3. The authors claimed that the fraction of mitochondrial proteins did not significantly change (Shown in Fig. 3B). Again, this could be due to cells at the pre-senescent stage; I wonder if this change could be significant after cells are fully senescent.

As discussed above, we provide now additional evidence demonstrating that cells are in a senescent state. Of note, our findings are in agreement with a recent report (PMID: 35987199), showing that mitochondrial and cellular proteomes are altered proportionally in senescent cells.

4. The data in Fig. 4E is not statistically significant. Please increase n to confirm your conclusion.

Considering the causal relationship between mitochondrial translation and OXPHOS deficiency, we feel that this point is rather confirmatory. We therefore present now instead two representative images (**updated Figure 4D**)

5. *Are these mitochondrial proteome alterations associated with the senescent cell's functional output or other features? Are these changes cell type-dependent.*

To address the functional impact of the mitochondrial reprogramming of metabolism in senescent cells, we blocked BCAA catabolism by knocking down the rate-limiting enzyme BCKDHA, before treating cells with decitabine or doxorubicin. However, the interpretation of these experiments was hampered by increased cell death upon BCKDHA knockdown independent of DNA damage. Thus, the BCKDHA appears to be essential for cell viability under our cell culture conditions, precluding an assessment of the role of the BCAA catabolism for the functional output of senescent cells. We included these data in the updated **Suppl. Figure 5D** and briefly mentioned them in the text on p. 10.

6. *I suggest the authors clarify explicitly the knowledge gap that the current study accomplished.*

We apologize for not being clear and have revised the manuscript to clarify this point. As pointed out in the General Statement, we have performed a comprehensive and time-resolved analysis of the mitochondrial proteome upon induction of cellular senescence and observed rapid metabolic rewiring of mitochondria. In particular, we identified two metabolic pathways, the 1C-folate metabolism and the BCAA catabolism that were rapidly rewired upon senescence induction. Our analysis therefore provides unprecedented insight into the function and metabolic adaptations of mitochondria in senescent cells, which could also serve as a reference for future studies. Moreover, we demonstrate that quantification of mitochondrial abundance using the mitochondrial volume allows to reconcile seemingly contradicting conclusions on the function/fitness of mitochondria in senescent cells. In our opinion, our studies therefore represent a valuable and significant contribution to our understanding of the role of mitochondria in senescence.

Minor: Please carefully check the statistical analysis. N of 2 is not sufficient for One-way ANOVA.

We have revisited the statistical analysis of our experiments and noted a mistake in the original **Figure 4E** (statistical analysis with n=2). We apologize for this mistake and have corrected it (see reply point 4) as updated Figure 4D.

Reviewer #2 (Significance (Required)):

*I do not see the significance of the current work. The work is descriptive and incremental, which reduce the impact of this manuscript.
My expertise is in cellular senescence and aging.*

As outlined in the General Statement and in reply to reviewer 2 (point 6), we provide insight into changes in the mitochondrial proteome in senescent fibroblasts in an unprecedented manner. Although metabolic reprogramming of mitochondria has been described, our results highlight the relevance of two metabolic pathways in senescent cells, the one-carbon and the BCAA metabolism, which to our knowledge has not been described before. Moreover, our work reveals the importance to adequately quantify mitochondrial abundance, when assessing the bioenergetic activity of mitochondria in (enlarged) senescent cells.

Reviewer #3 (Evidence, reproducibility and clarity (Required)):

The manuscript by Kim and colleagues investigated the metabolic and proteomic changes in senescent fibroblasts. The authors performed a time-resolved analysis of the proteome and revealed the impacts on mitochondrial proteome. They also applied tracing approaches to further demonstrate the impact on mitochondrial metabolism and revealed a potential impact of branched-chain amino acid catabolism and carbon-folate metabolism in senescent fibroblasts. They concluded that the reprogramming of mitochondria influences the senescence-associated secretory phenotype (SASP) impacting diseases associated with senescent cells.

1. The authors quantified the volume of mitochondria to examine mitochondrial functions. Some data were normalized "per cell" and "per mito". Should it read per mito volume? The comparison between the two normalization procedures in Fig 2 is important but also confusing. Can the authors speculate why non-mitochondrial respiration seems to be increased in decitabine/doxorubicin conditions (Fig. 2G/H)? Are the data significantly affected after normalization to non-mitochondrial respiration?

As anticipated by the reviewer, 'per mito' indeed refers to the mitochondrial volume. We have adjusted the text on p. 6 to clarify this point.

An increase in non-mitochondrial respiration in senescent cells was also reported by the von Zglinicki group (PMID: 28330601). A major source of non-mitochondrial OCR is thought to be cytosolic/peroxisomal H₂O₂ production by cyclooxygenase (COX), and NADPH oxidase, which consume oxygen to produce H₂O₂. Especially, the cytosolic H₂O₂ level is known to be higher in senescent cells (PMID: 10075689). Moreover, cyclooxygenase 1 (PTGS1) is significantly upregulated in our proteomics data (Reviewer figure 2), which could explain the reason for the increased non-mitochondrial OCR in senescent cells.

Reviewer figure 2. The pairwise comparison of PTGS1 level at each time point from the proteomics data in Figure 3. **: P < 0.01, ***: P < 0.001.

Of note, normalization to non-mitochondrial respiration does not significantly alter our conclusions. We observed a significant decrease in basal respiration and spare respiratory capacity both upon decitabine and doxorubicin treatment, which recapitulates our original findings taking mitochondrial abundance into account (**Reviewer figure 3**).

Reviewer figure 3. Oxygen consumption rates of senescent fibroblasts after normalization to non-mitochondrial OCR. (A, B) Left: representative graphs of OCR data. Right: quantification of respiratory parameters based on the OCR graph. Welch t-test, Bonferroni-Dunn correction, n=5.

2. The tracing data are impactful and critical to confirm metabolic changes. Can the authors explain why they added the tracer to the regular growth medium (instead of substituting the

metabolite of interest)? The media composition of the proteomic and the metabolic experiments is not identical. For instance, additional 5.5mM ¹³C glucose has been added to the ¹³C glucose tracer experiment while all other experiments were performed with 5.5mM ¹²C glucose only. Changes in media composition certainly affect cell function and metabolism. The authors may want to repeat key tracing experiments to mimic experimental conditions used in proteomics analysis.

The use of glucose-containing MEM had only practical reasons, because glucose-free MEM is not commercially available. However, this does not confound our proteomic analysis between day 1 and 7, since the tracing experiments in MEM containing glucose and ¹³C glucose were performed to the established senescent cells on day 7.

3. The impact on branched-chain amino acids is interesting. Did the authors observe ¹³C incorporation into the TCA cycle from BCAA? Increased BCAA catabolism may increase mitochondrial respiration, but the authors observed decreased OCR in senescent cells. Further, does inhibition of BCAA catabolism rescue the phenotype observed in senescent cells?

We did not observe any significant incorporation of BCAA carbons into malate but found an 3-5 fold increased flux into lipogenic TCA cycle intermediates, such as acetyl-CoA and citrate (updated Figure 5D). In agreement with glucose and glutamine being the major source of lipogenic acetyl-CoAs in cell culture (PMID: 31119666), the absolute changes are rather minor. We therefore carefully discuss the possibility that BCAA carbons are metabolized to supply the lipogenic carbon moieties in senescent cells rather than increasing mitochondrial respiration in the text on p. 12.

To address the functional impact of the mitochondrial reprogramming of metabolism in senescent cells, we blocked BCAA catabolism by knocking down the rate-limiting enzyme BCKDHA, before treating cells with decitabine or doxorubicin (see also Reviewer 2, point 5). However, the interpretation of these experiments was hampered by increased cell death upon BCKDHA knockdown independent of DNA damage. Thus, the BCKDHA appears to be essential for the cell viability under our cell culture conditions, precluding an assessment of the role of the BCAA catabolism for the functional output of senescent cells. We included these data in the updated Suppl. Figure 5D and briefly mentioned them in the text on **p. 10**.

Reviewer #3 (Significance (Required)):

Understanding the metabolic reprogramming of mitochondria in senescent cells is interesting and of high interest to the research community. However, some clarification is needed on experimental conditions, as media compositions in proteome and metabolome experiments were different which certainly affects cell metabolism and function.

Full Revision

We thank the reviewer for their positive evaluation of the significance of our work. In the revised manuscript, we have now clarified the technical issues raised showing that different culture conditions do not confound our proteomic and metabolomic analysis.

May 31, 2023

RE: Life Science Alliance Manuscript #LSA-2023-02127

Prof. Thomas Langer
Max-Planck-Institute for Biology of Ageing
Mitochondrial Proteostasis
Joseph-Stelzmann-Str. 9b
Cologne 50931
Germany

Dear Dr. Langer,

Thank you for submitting your revised manuscript entitled "Metabolic rewiring of mitochondria in senescence revealed by time-resolved analysis of the mitochondrial proteome". We would be happy to publish your paper in Life Science Alliance pending final revisions necessary to meet our formatting guidelines.

- please upload your main manuscript text as an editable doc file
- Please upload all figure files as individual ones, including the supplementary figure files; all figure legends should only appear in the main manuscript file
- please add a Running Title and a Summary Blurb/Alternate Abstract to our system
- please add a Category for your manuscript in our system
- please add the Twitter handle of your host institute/organization as well as your own or/and one of the authors in our system
- please add an Author Contributions section to your main manuscript text and in the system
- please add a conflict of interest statement to your main manuscript text
- please add your main and supplementary figure legends to the main manuscript text after the references section
- please revise the figure legend for Figure S1 such that the figure panels are introduced in alphabetical order
- please add a callout for Figure 2B-D to your main manuscript text

A. FINAL FILES:

B. MANUSCRIPT ORGANIZATION AND FORMATTING:

Sincerely,

Reviewer #1 (Comments to the Authors (Required)):

The authors have addressed all of my previous concerns. They performed the requested experiments and provided additional data that further support their findings.

Reviewer #2 (Comments to the Authors (Required)):

The revised manuscript is much improved over the previous version. I recommend its publication in your journal.

June 8, 2023

RE: Life Science Alliance Manuscript #LSA-2023-02127R

Prof. Thomas Langer
Max Planck Institute for Biology of Ageing
Mitochondrial Proteostasis
Joseph-Stelzmann-Str. 9b
Cologne 50931
Germany

Dear Dr. Langer,

Thank you for submitting your Research Article entitled "Time-resolved proteomic analyses of senescence highlights rewiring of mitochondrial metabolism". It is a pleasure to let you know that your manuscript is now accepted for publication in Life Science Alliance. Congratulations on this interesting work.

DISTRIBUTION OF MATERIALS:

Again, congratulations on a very nice paper. I hope you found the review process to be constructive and are pleased with how the manuscript was handled editorially. We look forward to future exciting submissions from your lab.

Sincerely,
